# Trajectories of plasma and CSF MTBR-tau243 and phosphorylated-tau species across the Alzheimer's disease continuum

Lyduine E. Collij [1,2,3] ✉, Gemma Salvadó [1], Kanta Horie [4,5,6], Nicolas R. Barthélemy [4,5], Tobey J. Betthauser [7,8], Olof Strandberg[1], Ruben Smith [1,9], Sebastian Palmqvist[1,9], Suzanne E. Schindler [5,10], Rik Ossenkoppele [1,11,12], Shorena Janelidze [1], Niklas Mattsson-Carlgren [1,9,13], Randall J. Bateman [4,5] & Oskar Hansson [1] ✉

To efficiently implement plasma and cerebrospinal fluid (CSF) biomarkers for staging and prognosis of Alzheimer disease (AD), we must understand their dynamics across disease progression. We analyzed participants from the Swedish BioFINDER-2 study with mass spectrometry measurements of plasma and CSF tau species, including eMTBR-tau243/MTBR-tau243 and phosphorylation occupancies (%p-tau). Disease duration was estimated using Aβ-PET and tau-PET with the SILA algorithm. Bootstrapped LOESS models showed that %p-tau217 changes earliest, increasing just before Aβ-PET positivity. Other p-tau species changed later, with smaller dynamic ranges and earlier ceiling effects. %p-tau205 and MTBR-tau243 changes aligned with tau-PET positivity onset, while MTBR-tau243—especially plasma eMTBR-tau243—tracked cortical tau burden in later stages. Non-phosphorylated mid-region tau may serve as a late-stage biomarker. Taken together, concurrent assessments of plasma or CSF %p-tau217, %p-tau205, and (e)MTBR-tau243 provides information about different biological events in the disease cascade, which can benefit clinical trials and patient management in clinical practice.

Recent developments in Alzheimer's disease (AD) clinical trials emphasize the need for accurate quantification of an individual's burden of pathology[1,2] to optimize the diagnostic work-up[3], decision-making regarding treatments[4], and stratification of prognosis[5–7]. Positron Emission Tomography (PET) is commonly used to assess in vivo brain burden of aggregates of amyloid-β (Aβ) and tau, the two pathological hallmarks of AD[8]. However, PET is limited by the low availability of scanners and high costs in most settings, which reduces its suitability for large-scale use. The ability to measure phosphorylated tau (p-tau) species at different residues in plasma or cerebrospinal fluid (CSF) has enabled more cost-effective disease tracking, supporting patient selection and AD disease staging[9–11].

[1]Clinical Memory Research Unit, Department of Clinical Sciences Malmö, Faculty of Medicine, Lund University, Lund, Sweden. [2]Radiology and Nuclear Medicine, Amsterdam UMC, location VUmc, Amsterdam, The Netherlands. [3]Brain Imaging, Amsterdam Neuroscience, Amsterdam, The Netherlands. [4]Tracy Family Stable Isotope Labeling Quantitation (SILQ) Center, Washington University School of Medicine, St. Louis, MO, USA. [5]Department of Neurology, Washington University School of Medicine, St. Louis, MO, USA. [6]Eisai, Inc., Nutley, NJ, USA. [7]Wisconsin Alzheimer's Disease Research Center, School of Medicine and Public Health, University of Wisconsin-Madison, Madison, WI, USA. [8]Department of Medicine, School of Medicine and Public Health, University of Wisconsin-Madison, Madison, WI, USA. [9]Memory Clinic, Skåne University Hospital, Malmö, Sweden. [10]Charles F. and Joanne Knight Alzheimer Disease Research Center, Washington University School of Medicine, St. Louis, MO, USA. [11]Neurology, Alzheimercenter Amsterdam, Amsterdam UMC, location VUmc, Amsterdam, The Netherlands. [12]Neurodegeneration, Amsterdam Neuroscience, Amsterdam, The Netherlands. [13]Wallenberg Center for Molecular Medicine, Lund University, Lund, Sweden. ✉e-mail: lyduine.collij@med.lu.se; oskar.hansson@med.lu.se

Plasma or CSF p-tau217 has been proposed as a particularly high-performing biomarker[12–16], with excellent accuracy in discriminating between Aβ-negative and Aβ-positive individuals and low-intermediate vs high tau-PET burden[17] and strong associations with AD disease progression in both preclinical and symptomatic stages[18,19]. In contrast, levels of the microtubule-binding region of tau (MTBR-tau243)[10] and potentially p-tau205[11] in CSF have been proposed as candidate biomarkers that reflect tau-PET burden. It remains to be determined if the recently developed plasma endogenously cleaved MTBR-tau243 (eMTBR-tau243) biomarker shows similar associations with tau PET[20].

Understanding changes in CSF and plasma tau species along the full AD pathological continuum is key to support implementation of these biomarkers in trials and clinical settings. In autosomal dominant AD, p-tau217 was shown to demonstrate the earliest change against estimated year to symptom onset (EYO), nearly two decades, closely followed by p-tau181 and years later by p-tau205[21]. Using different PET-based approaches to create relevant disease time scales, such as the sampled iterative local approximation (SILA) algorithm[22] or generalized additive model (GAM) approaches[23], similar trajectories were described for CSF and plasma p-tau biomarkers in sporadic AD, though the number of investigated species was limited and MTBR-tau243/eMTBR-tau243 was not investigated[23,24]. Also, it remains unclear how non-phosphorylated tau species change across the AD pathological continuum and their potential clinical value. Only one study to date showed that non-phosphorylated mid-region tau could be a valuable late-stage biomarker[9]. Finally, the potential effect of APOE-Ɛ4 carriership and biological sex on these fluid biomarker trajectories against PET time scales has not yet been investigated.

The aim of this study was therefore to investigate the temporal onset of abnormality and trajectories of mass spectrometry-based cross-sectional measurements of MTBR-tau243 and p-tau species in both CSF and plasma against disease time in relation to Aβ-PET and tau-PET positivity, as determined with the SILA algorithm[22]. In addition, we assessed how these trajectories were influenced by the behavior of the non-phosphorylated tau species and whether they were affected by APOE-Ɛ4 carriership and biological sex. Finally, longitudinal CSF data were available to investigate changes in biomarker slope against disease time. To this end, we included BioFINDER-2 participants across the clinical continuum with available CSF MTBR-tau243 and p-tau and plasma eMTBR-tau243 and p-tau measurements.

## Results

### Demographics
The CSF cohort ($n = 446$) consisted of 183 cognitively unimpaired participants and 263 cognitively impaired patients (mild cognitive impairment [MCI]: $n = 124$; dementia: $n = 139$), of which 219 (49.1%) were females, 258 (57.8%) were APOE-Ɛ4 carriers, and the mean age at baseline was 71.4 ( ± 8.50) years old (Table 1). Aβ-PET was available for 261 (58.5%) non-demented subjects, while tau-PET was available for all participants.

The plasma cohort ($n = 784$) consisted of 331 cognitively normal and 453 impaired patients (MCI $n = 245$; dementia: $n = 208$), of which 398 (50.8%) were females, 477 (60.8%) were APOE-Ɛ4 carriers, with a mean age of 72.2 ( ± 9.27) years (Table 1). Aβ-PET was available for 576 (73.5%) non-demented subjects, while tau-PET was available for all participants.

### PET-chronology measure
To retrieve an individual's disease duration, the estimated time of amyloid and tau PET positivity at the visit closest to the fluid biomarker was determined by applying the previously developed sampled iterative local approximation (SILA) algorithm to the whole BioFINDER-2 PET dataset. This dataset included Aβ-PET of 1408 individuals, of which 686 were longitudinal, with an average of 2.33 scans and mean follow-up time 2.96 ± 1.04 years and tau-PET of 2003 individuals, of which 922 had longitudinal data, with an average of 2.35 scans and a mean follow-up time 2.83 ± 1.07 (Supplementary Fig. 1)[22]. Global Aβ-PET burden was expressed in Centiloid (CL) units using the standard target mask available on the GAAIN website, while tau-PET burden was expressed in standard uptake value ratios (SUVRs) in a temporal meta-ROI, reflecting Braak I-IV (see "Methods"). The algorithm uses discrete sampling of CL/SUVR for Aβ/tau-PET versus age data to establish the relationship between CL/SUVR rate and CL/SUVR. Numerical smoothing (robust LOESS) and Euler's method are used to numerically integrate these data to generate a non-parametric CL/SUVR versus time curve. To give the integrated timeline meaning, the SILA algorithm sets time equal to zero to a user-specified value (i.e., threshold, "tipping point"), which was set at 20 CL[2] and 1.36[25] SUVR for Aβ-PET and tau-PET, respectively, to demarcate the zero time corresponding to the A+ and T+ threshold, respectively. The estimated years from biomarker positivity is calculated for each person by first solving this curve for time using a person's observed CL/SUVR, and subtracting the estimated A+ duration from their age at that scan[22]. This amyloid/tau "chronology" can be interpreted as the time from PET-detectable Aβ/tau accumulation and serves as the main outcome in the current work.

The SILA algorithm is freely available at GitHub (https://github.com/Betthauser-Neuro-Lab/SILA-AD-Biomarker).

### CSF biomarker measures over the course of the Aβ-PET chronology
Log-transformed and z-scored (compared to CU Aβ-negative group: $n = 97$) CSF Aβ42/40, %p-tau species (i.e., phosphorylated tau species occupancy or p-tau/non-p-tau*100), and MTBR-tau243 as a function of estimated years from Aβ-positivity onset or "chronology" are shown in Fig. 1A–C and Supplementary Figs. 2 and 3A. First, bootstrapped LOESS models suggested that CSF Aβ$_{42/40}$ starts to increase ~10 years before Aβ-PET positivity, followed by all %p-tau species, except %p-tau199 and %p-tau202. The only tau biomarker that reached significant abnormality (i.e., reaching >1.96 z-score compared to the reference population) before Aβ-PET positivity was %p-tau217, which became abnormal 0.6 year (95% CI −1.7 to 0.4) prior to Aβ-PET positivity (Fig. 1C).

While visual inspection of the %p-tau111, %p-tau153, %p-tau181, %p-tau208, and %p-tau231 curves suggested a plateau 10 years after Aβ-positivity, %p-tau217 demonstrated continued increases also after the onset of Aβ-PET abnormality. The biomarkers that most clearly changed only after Aβ-positivity were %p-tau205 (5.5 years, 95% CI 3.8–7.3) and MTBR-tau243 (6.7 years, 95% CI 5.2–8.3), with the latter showing the steepest curve closely following tau-PET burden (Fig. 1C). Finally, non-phosphorylated mid-region tau only reached significant abnormality upon advanced Aβ-PET disease duration (18.6 years, 95% CI 14.9–22.6), similarly to abnormality in global cognition, as measured by the modified Preclinical Alzheimer Cognitive Composite (mPACC).

### CSF biomarker measures over the course of tau-PET chronology
Log-transformed and z-scored CSF Aβ42/40, %p-tau species, and MTBR-tau243 as a function of estimated years from tau-positivity duration or "chronology" are shown in Fig. 1D–F and Supplementary Figs. 3B and 4. The pseudo-temporal ordering of biomarkers was similar to Aβ-PET chronology models, with CSF Aβ$_{42/40}$ first showing increased abnormality, followed by most %p-tau species, and finally MTBR-tau243 and %p-tau205. Bootstrapped LOESS models suggested that all biomarkers became significantly abnormal before tau-PET onset, except for the non-phosphorylated mid-region tau marker (6.4 years, 95% CI 4.6–8.3, Fig. 1F). MTBR-tau243 and %p-tau205 became abnormal closest to tau-PET onset, with significant increases less than 2 years before tau-PET burden in the temporal meta-ROI (MTBR-tau243:

## Table-1 | Demographics

| CSF cohort | | | | | | | |
|---|---|---|---|---|---|---|---|
| | CU Aβ- (N = 97) | CU Aβ + (N = 86) | MCI Aβ + (N = 101) | Dementia Aβ + (N = 115) | MCI Aβ- (N = 23) | Dementia Aβ- (N = 24) | Overall (N = 446) |
| Age, y | 69.6 (10.2) | 71.3 (9.25) | 72.1 (7.76) | 73.2 (6.68) | 71.8 (6.71) | 67.8 (8.91) | 71.4 (8.50) |
| Sex, F | 48 (49.5%) | 43 (50.0%) | 41 (40.6%) | 64 (55.7%) | 11 (47.8%) | 12 (50.0%) | 219 (49.1%) |
| APOE-ε4 carrier | 32 (33.0%) | 61 (70.9%) | 73 (72.3%) | 83 (72.2%) | 5 (21.7%) | 4 (16.7%) | 258 (57.8%) |
| PACC | −0.08 (0.77) | 0.21 (0.80) | 1.88 (0.89) | 4.32 (1.78) | 1.10 (0.71) | 2.91 (1.39) | 1.64 (2.02) |
| Centiloid | 0.94 (0.07) | 1.26 (0.26) | 1.47 (0.26) | 1.75 (0.21) | 0.91 (0.03) | 0.94 (0.17) | 1.24 (0.32) |
| missing | 18 (18.6%) | 7 (8.1%) | 13 (12.9%) | 107 (93.0%) | 18 (78.3%) | 22 (91.7%) | 185 (41.5%) |
| Aβ-PET chronology | n/a | 6.35 (4.36) | 9.51 (5.23) | 15.6 (5.41) | n/a | n/a | 8.53 (5.42) |
| Tau-PET temporal meta-ROI | 1.16 (0.0876) | 1.23 (0.203) | 1.49 (0.459) | 2.26 (0.666) | 1.14 (0.0771) | 1.18 (0.161) | 1.53 (0.611) |
| Tau-PET chronology | n/a | 2.03 (3.29) | 4.92 (3.87) | 8.30 (3.61) | n/a | n/a | 6.82 (4.19) |
| Plasma cohort | | | | | | | |
| | CU Aβ-(N = 121) | CU Aβ + (N = 210) | MCI Aβ + (N = 156) | Dementia Aβ + (N = 180) | MCI Aβ-(N = 89) | Dementia Aβ- (N = 28) | Overall (N = 784) |
| Age, y | 63.8 (11.6) | 73.9 (8.33) | 74.2 (7.20) | 74.6 (7.30) | 70.7 (8.85) | 74.7 (6.25) | 72.2 (9.27) |
| Sex, F | 70 (57.9%) | 110 (52.4%) | 70 (44.9%) | 109 (60.6%) | 34 (38.2%) | 5 (17.9%) | 398 (50.8%) |
| APOE-ε4 carrier | 55 (45.5%) | 151 (71.9%) | 117 (75.0%) | 125 (69.4%) | 22 (24.7%) | 7 (25.0%) | 477 (60.8%) |
| PACC | 0.12 (0.71) | −0.19 (0.72) | −1.65 (0.94) | −2.79 (2.00) | −1.34 (0.68) | −2.44 (1.49) | −1.30 (1.67) |
| Centiloid | −6.21 (7.86) | 49.7 (38.2) | 81.6 (38.2) | 81.3 (38.4) | −4.80 (9.83) | −2.10 (11.9) | 42.9 (47.7) |
| missing | 23 (19.0%) | 13 (6.2%) | 13 (8.3%) | 128 (71.1%) | 18 (20.2%) | 13 (46.4%) | 208 (26.5%) |
| Aβ-PET chronology* | n/a | 8.25 (4.81) | 12.2 (5.44) | 12.4 (5.51) | n/a | n/a | 10.2 (5.58) |
| Tau-PET temporal meta-ROI | 1.14 (0.0819) | 1.28 (0.296) | 1.54 (0.456) | 1.92 (0.679) | 1.17 (0.0959) | 1.21 (0.103) | 1.44 (0.508) |
| Tau-PET chronology* | n/a | 4.45 (3.88) | 4.90 (3.38) | 7.53 (4.19) | n/a | n/a | 6.02 (4.11) |

PACC Preclinical Alzheimer Cognitive Composite, n/a not available.
*Only shown for individuals who reached the threshold and have robust estimates.

−1.9 years, 95% CI = −2.7 to −0.8; %p-tau205: −1.6 years, 95% CI = −2.5 to −0.7). Finally, tau-PET positivity was closely related to reaching abnormality in global cognition (mPACC).

In line with the Aβ-PET chronology observation, visual inspection suggested that most %p-tau species showed a ceiling effect closely following tau-PET onset. Only %p-tau205, and even more prominently MTBR-tau243, showed continued increased abnormality after tau-PET onset. While %p-tau205 also demonstrated a ceiling effect upon longer tau-PET disease duration, MTBR-tau243 showed a remarkable continued increase in abnormality (z-score >12).

### Effect of biological sex and APOE-ε4 carriership on CSF trajectories

Stratified analysis illustrated that the general order of CSF biomarkers reaching significant abnormality against the reference population (i.e., reaching >1.96 z-score based on CU Aβ-negative individuals) was not affected by biological sex or APOE-ε4 carriership. Across all strata, %p-tau217 consistently emerged as the earliest biomarker to become abnormal, %-ptau205 and MTBR-tau243 to closely align with tau-PET onset, and finally np-tau mid-region showing abnormality late in the disease course. In addition to the preserved ordering, we observed that APOE-ε4 carriers reached significant abnormality for all CSF biomarkers earlier than non-carriers, relative to both Aβ-PET and tau-PET reference onsets. Based on the 95% confidence intervals, this difference was only significant for %-ptau217 and %-ptau181 relative to Aβ-PET and tau-PET chronicity, and MTBR-tau243 relative to tau-PET chronicity. Conversely, global cognitive decline, assessed using the modified PACC, occurred later in APOE-ε4 carriers compared with non-carriers, though not significantly. Please note that confidence intervals were wider in stratified analyses due to the smaller sample sizes of the subpopulations, particularly for np-tau mid-region. Full results can be found in Supplementary Table 1.

### CSF dynamics and interindividual variability

As described above, for most %p-tau biomarkers, a ceiling effect in abnormality was observed ~10 years after Aβ-PET and 5 years after tau-PET onset, respectively. To understand the cause of this phenomenon, we investigated the trajectories of each p-tau species (i.e., %p-tau, p-tau, and non-phosphorylated tau [np-tau]) across chronology measures. For tau species that included residue 217, we observed that the %p-tau217 ceiling effect was explained by the increases in the reference epitope (non-phosphorylated mid-domain tau, np-tau212) at later AD pathological PET chronology (Fig. 2A). Similar effects were consistently observed across tau species (Supplementary Figs. 5 and 6).

To determine if MTBR-tau243 would track even more closely with tau-PET when correcting for interindividual variability, we corrected the biomarker for CSF dynamics with the closest p-tau reference epitope (i.e., np-tau212). Correcting MTBR-tau243 did not result in more similar trajectories to tau-PET burden in intermediate or late tau-PET regions (Braak III-IV/V-VI), rather this resulted in a more similar trajectory to tau-PET burden in early tau accumulating regions (Braak I–II, Fig. 2B).

### Longitudinal CSF p-tau species and MTBR-tau243

Longitudinal CSF data were available for 218 individuals, with a mean follow-up time of 2.04 years (SD = 0.21, range: 1.63–3.06 years). Similar to the cross-sectional analysis, CSF biomarker slopes of %p-tau species derived from linear mixed models showed a ceiling effect based on visual inspection shortly after amyloid- and tau-PET positivity onset, except for %p-tau205 (Fig. 3A, right figure and Supplementary Fig. 8). This can be explained by the higher accumulation rate of the phosphorylated form of 205 compared to the other species (Supplementary Fig. 8). Similarly, MTBR-tau243 demonstrated increased slopes upon high AD pathological PET chronology. In the case of amyloid-PET disease duration (Fig. 3A), this is most likely driven by insoluble tau burden, which is known to increase at higher levels of amyloid[7], and

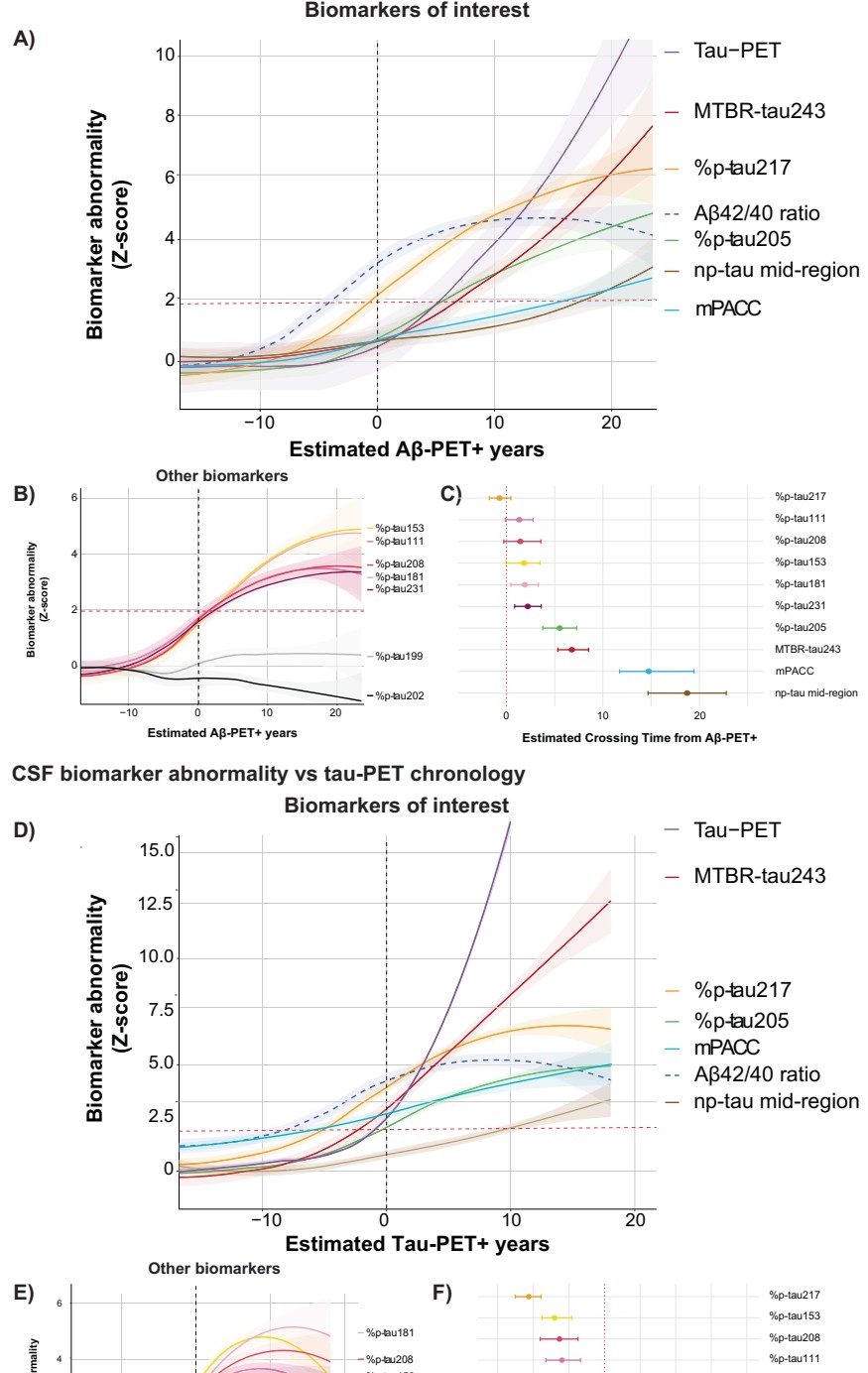

given the identical trajectory observed against tau-PET disease duration (Fig. 3B). The slope of MTBR-tau243 was reduced when adjusting the marker for the reference epitope np-tau212, mainly at high tau-PET disease duration. This is due to the increased accumulation of np-tau212 in individuals with higher tau burden (Fig. 3B). All reference np-tau variants showed highly similar accumulation patterns (Supplementary Fig. 8).

## Plasma biomarker measures over the course of the Aβ-PET chronology

We next investigated log-transformed and z-scored plasma eMTBR-tau243 and p-tau species biomarker trajectories as a function of estimated years from Aβ-PET positivity onset or "chronology" (Fig. 4 and Supplementary Fig. 9). Similarly to CSF, bootstrapped LOESS models suggested that plasma %p-tau217 showed significant

**Fig. 1 | CSF biomarker abnormality against estimated AD-pathological PET positivity in years.** CSF biomarker abnormality against **A**−**C** estimated Aβ-PET positivity in years, with onset (*x* axis = 0) set at CL = 20 and **D**−**F** estimated tau-PET positivity in years, with onset (*x* axis=0) set at temporal META-ROI = 1.36 SUVR. PET-positivity onset is indicated with the black dotted line. **A**, **B**, **D**, **E** Illustrate changes in z-scores, based on the mean and standard deviation of the cognitively unimpaired amyloid-negative cases. The solid line illustrates the mean population trajectory, while the shaded area's indicate the 95% confidence interval. Red dotted line for z-score figures illustrates the 1.96 SD from the reference population. **C**, **F** Forest plots demonstrate the point of p-tau, np-tau mid-region (np-tau212-221),

and MTBR-tau243 biomarkers crossing the Z-score=1.96 from the reference population, with 0 (red dotted line) indicating amyloid/tau-PET positivity, respectively. The 95% confidence intervals were retrieved from 1000 iteration bootstrapping (*n* = 261 and *n* = 446 for amyloid and tau, respectively). Note, biomarkers highlighted in panels A and D included those serving as reference standards (i.e., tau-PET burden and mPACC), identified as either the earliest (Aβ42/40) or latest changes (np-tau mid-region), or showing the largest increment (MTBR-tau243, %p-tau205, and %p-tau217) with increasing PET positivity time. Also, %p-tau199 and %p-tau202 are not included in the forest plot, as they did not reach sufficient biomarker abnormality at any point of positivity estimations amyloid-/tau-PET.

### A) CSF p-tau 217 species vs Aβ- and tau-PET positivity estimations

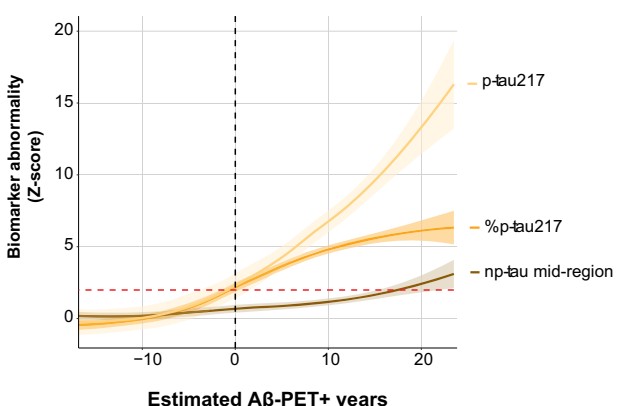 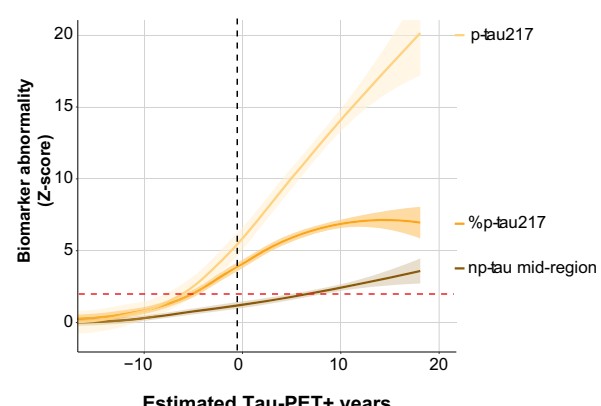

### B) MTBR-tau 243 species vs Braak regions

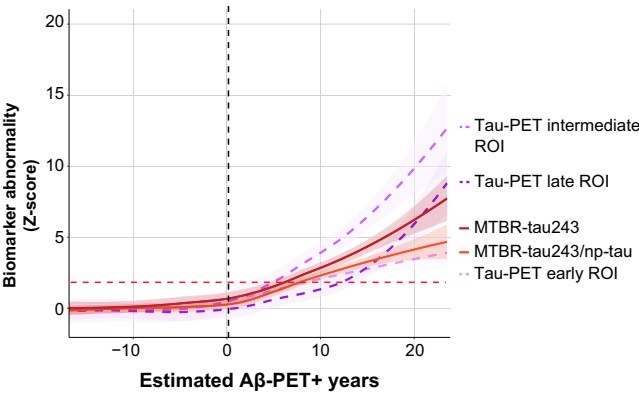 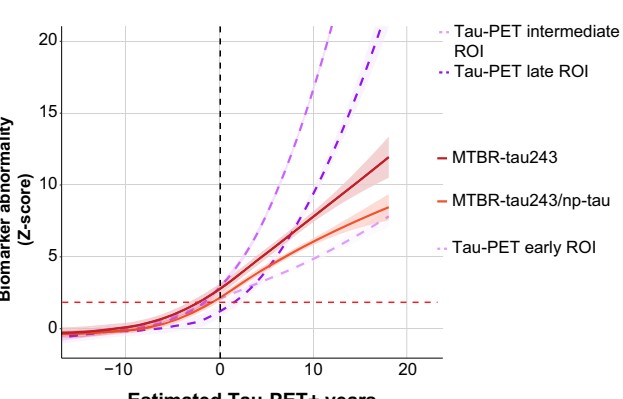

**Fig. 2 | CSF dynamics across estimated AD-pathological PET positivity in years.** CSF biomarker abnormality against estimated Aβ-PET positivity in years, with onset set at CL = 20 and estimated tau-PET positivity in years, with onset set at temporal META-ROI = 1.36 SUVR. PET-positivity onset is indicated with the black dotted line. **A** Illustrates changes in z-scored p-tau 217 forms (ratio: %p-tau217; phosphorylated p-tau: p-tau; and the non-phosphorylated reference epitope: np-tau 212–221), based on the mean and standard deviation of the cognitively unimpaired amyloid-negative cases. Shaded area's indicate the 95% confidence interval. While %p-tau217 plateaus at higher AD pathological burden, phosphorylated p-tau217 shows steep continued increase in abnormality. This effect is also apparent, thought to a lesser

extent, in the reference epitope np-tau212. **B** Illustrates the behavior of MTBR-tau243 and tau burden in PET regions-of-interest (ROIs) across chronology measures. Correcting MTBR-tau243 for CSF dynamics by means np-tau212 (the reference epitope of p-tau217) did not improve its association with intermediate and late tau ROIs, rather it resulted in a closer association with tau burden in early tau-accumulating (Braak I–II) regions. Red dotted line for z-score figures illustrates the 1.96 SD from the reference population. For all figures, the solid/dashed line illustrates the mean population trajectory, while the shaded area's indicate the 95% confidence interval.

abnormality already before Aβ-PET onset, becoming abnormal 1.8 years (95% CI −2.8 to −0.9) prior to Aβ-PET positivity. In addition, % p-tau181 and %p-tau205 also increased with disease severity, while %p-tau202 decreased with increased disease stage. %p-tau181 demonstrated a smaller dynamic range compared to %p-tau217, probably due to its higher interindividual variability

(Supplementary Figs. 10 and 11), reaching statistical abnormality only late in the disease course (14.7, 95% CI 11.7–18.3). Due to the small variability in plasma eMTBR-tau243 in the reference population (Supplementary Figs. 10 and 11), a significant abnormality compared to Aβ-PET onset could not be reliably estimated. Non-phosphorylated mid-region tau showed no changes before Aβ-PET

## A) Slopes CSF %p-tau species and MTBR-tau243 vs PET estimations

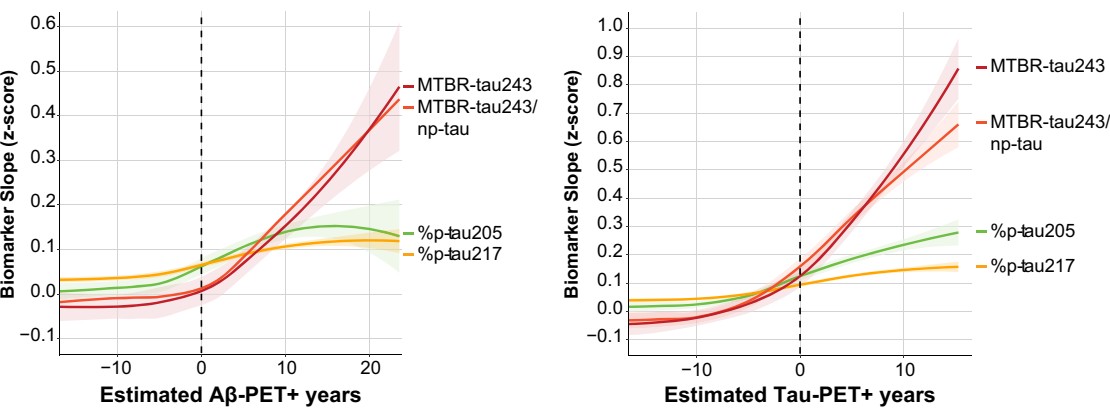

## B) Slopes CSF p-tau217 species vs Aβ- and tau-PET estimations

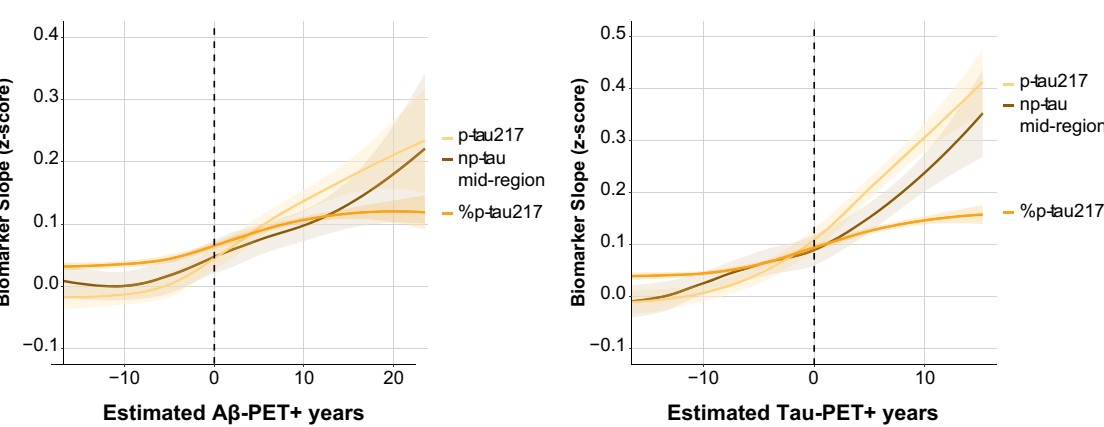

**Fig. 3 | Longitudinal CSF p-tau species and MTBR-tau243.** CSF biomarker abnormality against estimated Aβ-PET positivity in years, with onset set at CL = 20 and estimated tau-PET positivity in years, with onset set at temporal META-ROI = 1.36 SUVR. PET-positivity onset is indicated with the black dotted line. **A** Illustrates slope in key %p-tau species, MTBR-tau243 and MTBR-tau243 corrected for the non-phosphorylated reference epitope 212 (np-tau212) vs amyloid- and tau-PET chronology. **B** Illustrates the slopes of p-tau217 forms (ratio: %p-tau217; phosphorylated p-tau: p-tau; and the non-phosphorylated reference epitope: np-tau 212) across amyloid- and tau-PET chronology. For all figures, the solid line illustrates the mean population trajectory, while the shaded area's indicate the 95% confidence interval.

onset and did not reach significant abnormality during amyloid-based disease duration.

### Plasma biomarker measures over the course of the tau-PET chronology

Compared to estimated tau-PET disease duration, %p-tau217 demonstrated continued increases in abnormality many years after tau-PET positivity (Fig. 4D). This contrasts with its CSF counterpart, for which an attenuation effect was observed (Fig. 1D). In addition, eMTBR-tau243 and %p-tau205 became abnormal close to tau-PET onset (MTBR-tau243: −3.7 years, 95% CI = −4.5 to −2.5; %p-tau205: 1.9 years, 95% CI = 1.16−2.7) and showed clear increases subsequently. Specifically, the plasma eMTBR-tau243 trajectory closely followed the tau-PET burden trajectory, even more closely than CSF MTBR-tau243. Similar to CSF, the non-phosphorylated mid-region tau plasma marker demonstrated increases only at late-stage disease duration, reaching significantly abnormality late in the disease course (14.3 years, 95% CI 13.6−15.2) (Fig. 4C, D).

### Effect of biological sex and *APOE-ε4* carriership on plasma trajectories

Similar to CSF, stratified analysis illustrated that the general order of plasma biomarkers reaching significant abnormality against the reference population (i.e., reaching >1.96 z-score based on CU Aβ-negative individuals) was not affected by biological sex or *APOE-ε4* carriership. Across all strata, %p-tau217 consistently emerged as the earliest biomarker to become abnormal, followed by MTBR-tau243 and %-ptau205, and global cognitive decline abnormality late in the disease course based on years of tau-PET positivity. Note, np-tau mid-region onset could not be reliably estimated in the subpopulations. *APOE-ε4* carriers showed earlier changes in %-ptau217, %p-tau205, and mPACC compared to non-carriers, though not for eMTBR-tau243. Note, these differences did not reach statistical significance given the wider confidence intervals in these stratified analyses due to the smaller sample sizes of the subpopulations. Full results can be found in Supplementary Table 2.

### Plasma dynamics and interindividual variability

Finally, we investigated the dynamics of separate plasma p-tau species (i.e., %p-tau, p-tau, and np-tau) and the effect of interindividual variability correction on plasma eMTBR-tau243, using np-tau212. In contrast to the behavior of separate CSF species, plasma p-tau217 showed upon visual inspection less continued increase than the %p-tau217 form, which was consistently observed across tau species (Fig. 5A and Supplementary Figs. 12 and 13). Similar to CSF, np-tau212 showed

## Plasma biomarker abnormality vs PET estimates

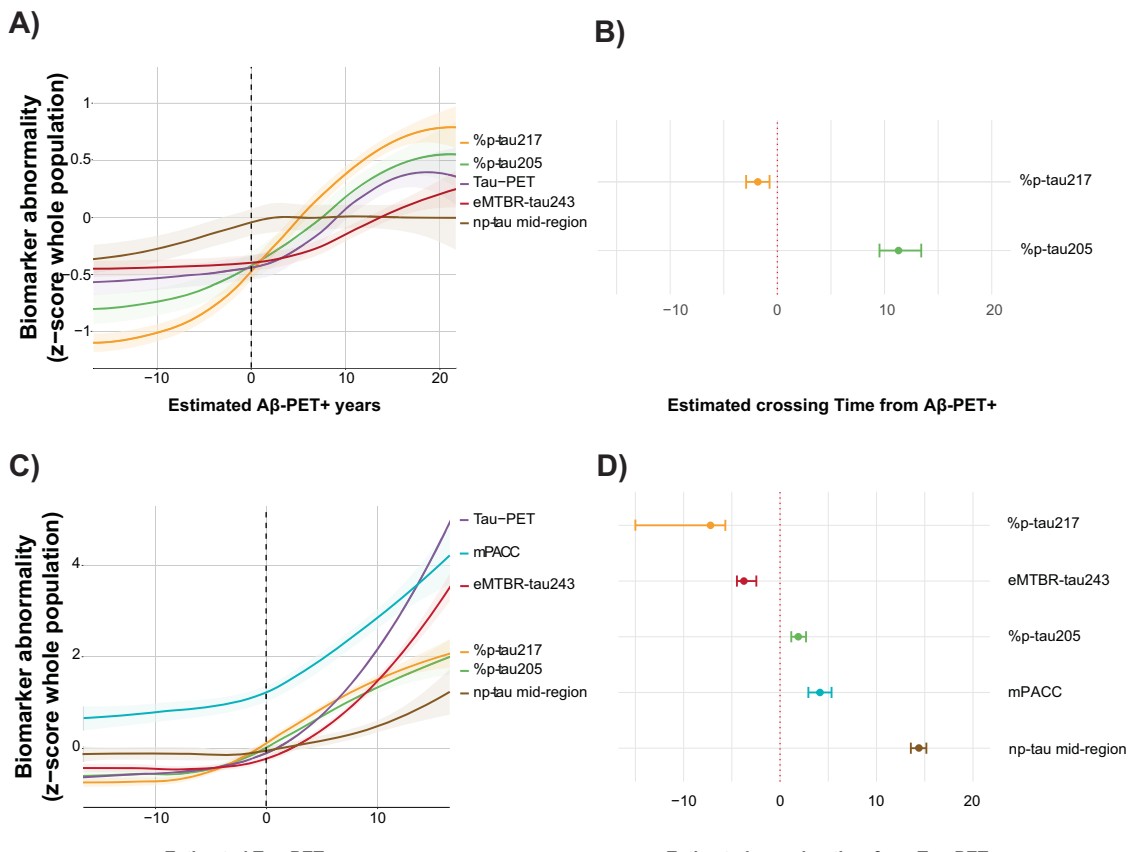

**Fig. 4 | Plasma biomarker abnormality vs estimated AD-pathological PET positivity.** Plasma biomarker abnormality against **A**, **B** estimated Aβ-PET positivity in years, with onset set at CL = 20, and **C**, **D** estimated tau-PET positivity in years, with onset set at temporal META-ROI = 1.36 SUVR. PET-positivity onset is indicated with the black dotted line. **A**, **C** Illustrate changes in z-scores, based on the mean and standard deviation of the whole population, rather than a CU Aβ- reference group, as plasma eMTBR-tau243 has essentially zero variance in CU Aβ-negative individuals. The solid lines represented the mean population trajectory, while the shaded area's indicate the 95% confidence interval. **B**, **D** Forest plots demonstrate

the point of p-tau, np-tau mid-region, and MTBR-tau243 biomarkers crossing the 1.96 SD from the reference population (different for each biomarker), with 0 (red dotted line) indicated amyloid/tau-PET positivity, respectively. The 95% confidence intervals were retrieved from 1000 iteration bootstrapping ($n = 576$ and $n = 784$ for amyloid and tau, respectively). Note that PACC and np-tau mid-region are not included in the amyloid-PET forest plot, as they did not reach sufficient biomarker abnormality at any point of amyloid-PET chronology. In addition, eMTBR-tau243 is also not included in the amyloid-PET forest plot due to unreliable estimates in this disease stage.

increases particularly at longer disease duration (Fig. 5A). Correcting plasma eMTBR-tau243 for np-tau212 showed no effect on its trajectory. Plasma eMTBR-tau243 trajectories closely follow tau-PET burden trajectories in intermediate and late tau-PET regions (Braak III–VI), but were less associated with early tau-PET burden (Braak I–II, Fig. 5B).

## Discussion

In a memory clinic population encompassing the full AD pathological continuum, we demonstrate that both CSF and plasma biomarkers show highly similar trajectories across AD-PET positivity duration and distinct temporal abnormality onset across species (Fig. 6). More specifically, %p-tau217 in both CSF and plasma showed significant abnormality just before Aβ-PET positivity and increased abnormality at longer disease duration, though less of a ceiling effect was observed for plasma %p-tau217 compared to CSF %p-tau217. In turn, %p-tau205 and (e)MTBR-tau243 demonstrated the most change after tau-PET positivity onset, with particularly plasma eMTBR-tau243 showing very similar trajectories as tau-PET burden in intermediate and late tau-PET regions. Other CSF and plasma %p-tau species showed highly similar trajectories to one another and a ceiling effect early on. The observed ceiling effect in the CSF ratios of p-tau species is due to the continued increase of the non-phosphorylated reference variants. In parallel, CSF

phosphorylated p-tau species showed continued increases upon longer amyloid and tau PET disease duration. This effect was less apparent for plasma measures. These findings contribute to our understanding of the trajectories of tau biomarkers in CSF and plasma throughout AD progression, and particularly their timing of abnormality onset, which is key to optimizing their use in clinical practice and clinical trials.

MTBR-tau243 has previously been shown to correlate strongly with tau-PET burden and cognitive measurements[10,20,26]. As such, it has been proposed as an insoluble tau aggregate biomarker for the revised A/T/(N) criteria[10] and for tau-directed therapies[26]. This study uniquely investigated the trajectories of CSF and plasma (e)MTBR-tau243 against approximated AD disease duration as measured with PET using a validated model[22]. Here we report that CSF MTBR-tau243 shows significant changes approximately 2 years before tau-PET positivity onset and continued increase in abnormality with longer disease duration, in contrast to all %p-tau species (e.g., MTBR-tau243 z-score$_{max}$ = 18.2 vs. %p-tau217 z-score$_{max}$ = 8.9). Interestingly, while plasma eMTBR-tau243 showed highly similar trajectories, its trajectories mirrored cortical tau-PET burden even more strongly compared to its CSF counterpart. In line with previous work[10], both CSF and plasma (e)MTBR-tau243 showed no significant change before Aβ-PET

## A) Plasma p-tau species vs Aβ- and tau-PET estimates

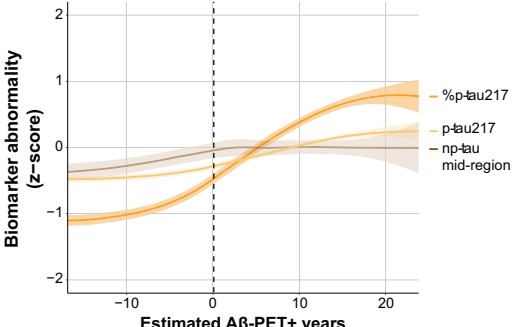
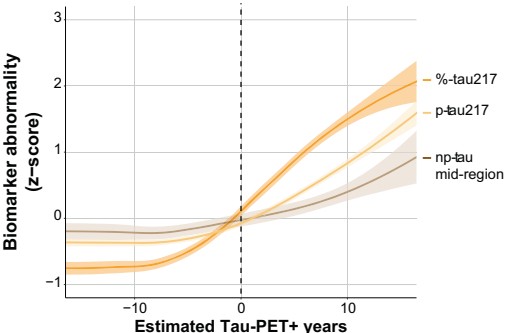

## B) Plasma eMTBR-tau243 species vs Braak regions

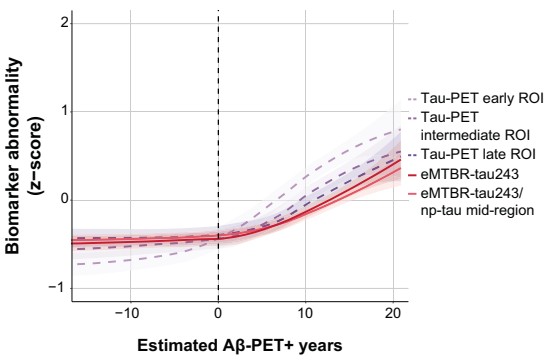
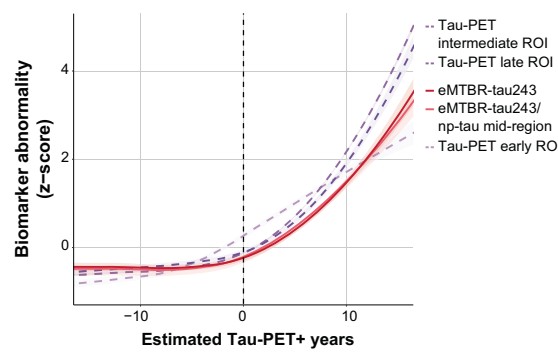

**Fig. 5 | Plasma dynamics across estimated AD pathological PET positivity.**
**A** Illustrates changes in z-scored p-tau 217 species (ratio: %p-tau217; phosphory-lated p-tau: p-tau; and the non-phosphorylated reference epitope: np-tau 212), against estimated amyloid- and tau-PET positivity in years, respectively. **B** Illustrates the behavior of MTBR-tau243 and tau burden in 3 regions-of-interest (ROIs) across

chronology measures. Correcting MTBR-tau243 for plasma dynamics by means of np-tau212 (the reference epitope of p-tau217) showed minimal to no effect. PET-positivity onset is indicated with the black dotted line. For all figures, the solid/dashed line illustrates the mean population trajectory, while the shaded area's indicate the 95% confidence interval.

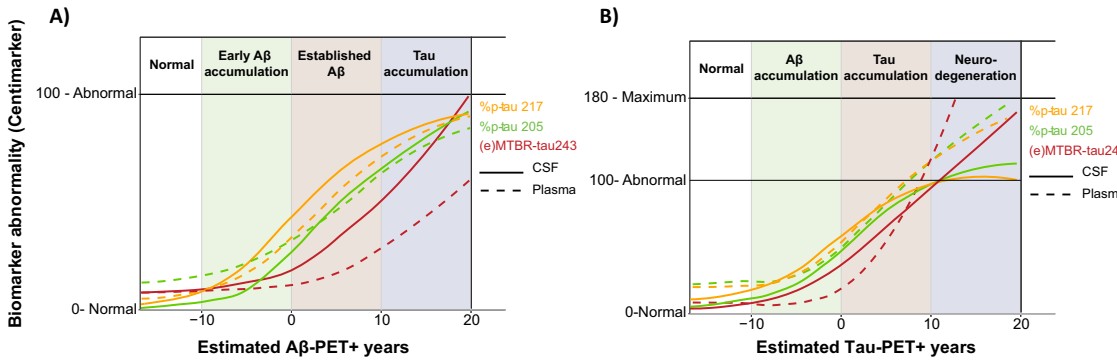

**Fig. 6 | Trajectories of core CSF and plasma biomarkers in Centimarker.** The figure illustrates the trajectories of core biomarkers (%p-tau205, %p-tau217, CSF MTBR-tau243, and plasma eMTBR-tau243) in CSF (solid lines) and plasma (dashed lines) using the Centimarker metric across estimated **A** Aβ-PET and **B** tau-PET positivity in years. Centimarker measure was anchored at the mean of cognitively unimpaired amyloid-negative individuals (=0) and amyloid- and tau-positive

dementia patients (=100). Highly similar trajectories between CSF and plasma tau biomarkers can be appreciated, with plasma closely following changes in CSF, though showing less of a ceiling effect upon longer disease duration. Note that CSF and plasma trajectories were derived based on their respective cohorts, which had only some overlap in individuals.

onset, further supporting the tau-specificity of this biomarker and its potential for disease staging and risk-stratification efforts.

We also investigated the behavior of several mass spectrometry-based CSF and plasma p-tau species and species across PET disease duration. We consistently observed that both CSF and plasma %p-tau217 showed the most reliable early change, -1–2 years before Aβ-

PET onset. Interestingly, plasma %p-tau217 demonstrated less of a ceiling effect upon longer disease duration compared to CSF %p-tau217. This suggests that CSF measures might be more susceptible to disease-associated changes in protein metabolism and transport compared to plasma biomarkers[27]. In line with previous studies[11,28], %p-tau205 showed closer coupling to tau-PET disease onset and burden

than other p-tau species, though still to a lesser extent than MTBR-tau243. P-tau species other than p-tau217 and p-tau205 showed significant change later in the disease course, smaller dynamic ranges, and early ceiling effects. For example, %p-tau217 outperformed %p-tau181 for both CSF and plasma biomarkers, probably due to the larger variability of the latter and the previously described durability of the former. More specifically, previous work has illustrated that %-ptau217 is more resistant against intracellular cleavage, meaning that even a modest early increase in the molecule can accumulate into a detectable signal[29]. These findings are also in line with previous literature, which reported optimal performance of %p-tau217 to predict Aβ-status[13,15,28]. In contrast to previous studies, we did not observe an early change of CSF %p-tau231 against Aβ-PET onset[30–33]. Rather, it performed similarly to CSF %p-tau111, %p-tau153, %p-tau181, and %p-tau208, in line with previous work[33]. Unfortunately, plasma %p-tau231 was not available, and it would be of interest to determine the performance of this biomarker.

Our stratified analysis suggests that *APOE-Ɛ4* carriers demonstrated changes in both CSF and plasma biomarkers earlier than non-carriers, though not a shorter disease trajectory per se. As the SILA model is anchored in the same positivity cut-point for both subpopulations, and previous work has illustrated that the rate of Aβ-PET accumulation is not affected by *APOE-Ɛ4* carriership[22], this difference is not driven by the presence of higher aggregated Aβ, as measured by PET, in *APOE-Ɛ4* carriers. Instead, a potential underlying biological mechanism could be that fluid and PET Aβ provide partially independent information[34] and *APOE-Ɛ4* seems to affect the balance between soluble and aggregated Aβ forms. In particular, *APOE-Ɛ4* was found to promote the accumulation of amorphous Aβ assemblies, which are more reflected in the fluid biomarkers compared to the PET imaging[35]. In line, Mastenbroek et al. reported that homozygous *APOE-Ɛ4* carriers had relatively more soluble Aβ compared to fibrillar Aβ[36]. Together with the observation that most p-tau forms reflect both Aβ and tau pathology[37], it is possible that this shift in biomarker abnormality onset for *APOE-Ɛ4* carriers reflects the relatively higher presence of soluble pathological markers in this subpopulation, making them potentially more sensitive for early pathological detection than in *APOE-Ɛ4* non-carriers. Given that these analyses were performed in subpopulations, resulting in much larger confidence intervals, these findings should be interpreted with caution and be validated in an external dataset. Related, due to cohort size, we were not able to investigate the effect of *APOE-Ɛ4* within males and females separately. As previous studies reported faster disease progression for particularly female *APOE-Ɛ4* carriers[38], it would be of interest to investigate the potential interplay between *APOE-Ɛ4* carriership and biological sex on biomarker disease trajectories.

In the current work, we used mass spectrometry-based p-tau occupancy measures, which are generally preferred due to their high specificity[11]. This method enables simultaneous measurement of CSF/plasma np-tau and p-tau species, effectively providing ratio measures[11] and accounting for interindividual variability[27]. However, as described above, most of these biomarkers demonstrated a ceiling effect after Aβ-PET onset. To understand this phenomenon, we investigated the evolution of np-tau and p-tau species separately across the AD pathological continuum, as well as their longitudinal trajectories in CSF. For CSF species, we observed that both the phosphorylated and non-phosphorylated species of all p-tau species showed continuous increase in abnormality, causing a ceiling effect in their ratios in the latest stages. While this effect was less apparent for raw plasma phosphorylated species across p-tau species, again the non-phosphorylated species showed increases mainly upon longer disease duration. Thus, plasma non-phosphorylated mid-region tau has the potential of being a late-stage biomarker, similar to CSF[9].

Our work comes with some methodological considerations. First, though the CSF cohort was of acceptable size, it was considerably

smaller compared to the plasma cohort. In addition, while CSF data were collected at baseline for all participants, plasma data were collected at different study visits. As a consequence, the overlap between the two cohorts was minimal, limiting statistical comparisons of CSF- and plasma-based trajectories. Second, while the SILA algorithm used all available longitudinal PET data to fit the time estimations, only the cross-sectional estimated disease duration closest to the fluid biomarker was used. In addition, negative SILA estimations further away from the cut-point should be treated with caution, as model fit is suboptimal due to the noise characteristics of PET imaging. Third, this study utilized the radiotracers [18F]flutemetamol and [18F]RO948 for the assessment of Aβ and tau burden, respectively. Though amyloid-PET was harmonized using the Centiloid approach and our model fit was highly consistent with previous reports using the SILA method on other F-18 radiotracers, such as [18F]florbetapir[39], harmonization for tau-PET tracers is still not established, and differences in tau-PET tracer kinetics might affect their association with fluid biomarker trajectories. Nonetheless, a previous head-to-head study from our group has shown highly comparable performance between [18F]RO948 and the FDA-approved [18F]flortaucipir tracer[40]. Fourth, as per the BioFINDER-2 study design, Aβ-PET was only available in very few patients with a dementia diagnosis at baseline, though the whole range of Aβ burden was well represented due to the inclusion of individuals with MCI and did not affect SILA model fit. A strength of the study was the availability of a relatively large longitudinal PET dataset ($N_{Aβ} = 686$ and $N_{tau} = 922$) compared to previous studies[22,39] with an average follow-up time of nearly 3 years for both PET tracers, which supports accurate model fit by the SILA algorithm. Fifth, this study utilized a previously validated and commonly used temporal meta-ROI to assess global tau-PET burden, though known heterogeneity in tau-PET exists[41]. As such, it would be of interest to investigate whether the investigated fluid biomarkers in this work demonstrate distinct trajectories based on previously described tau-PET subtypes. Sixth, the SILA algorithm assumes individuals follow consistent accumulation trajectories, which could be a limitation. Nonetheless, this assumption has been extensively tested for amyloid-PET in sporadic AD, showing no effect of sex, *APOE* genotype, age, and clinical diagnosis on trajectories[22]. Similar work is currently also ongoing for tau-PET. Seventh, as plasma eMTBR-tau243 has essentially zero variance in CU Aβ-negative individuals, the estimated crossing to abnormality based on this reference population should be interpreted with caution. For this biomarker, a z-score>1.96 was easily achieved and often before Aβ-PET onset, as reflected in the large confidence interval in the forest plot. Nonetheless, the biomarker trajectories normalized based on the whole plasma cohort demonstrate that eMTBR-tau243 does not show a significant change before Aβ-PET onset. Eighth, longitudinal plasma tau biomarkers were not available. Future work should investigate the stability of these markers over time. Finally, while the BioFINDER-2 cohort consists of a memory clinic population, it may not be broadly generalizable, and future studies should include larger, more real-world populations, including cohorts with a higher frequency of other neurodegenerative or psychiatric diseases, medical comorbidities, and other demographic backgrounds that may affect biomarker, and particularly plasma biomarker, measurements.

Taken together, in a population encompassing the full AD pathological continuum, we observed that CSF and particularly plasma (e)MTBR-tau243 trajectories were closely associated with trajectories of cortical tau-PET burden (Braak III–VI). In addition, %p-tau217 was the only tau biomarker that demonstrated reliable changes before Aβ-PET positivity onset, while %p-tau205 was more closely associated with tau-PET positivity onset. Finally, both CSF and plasma non-phosphorylated mid-region tau could be a potential late-stage biomarker. Taken together, concurrent assessment of (e)MTBR-tau243, p-tau217, p-tau205, and potentially phosphorylated mid-region tau can accurately track PET-based AD pathological burden. This can support

staging efforts at the individual level using CSF or plasma measurements, which is an increasing need for clinical trial design. In addition, combining different p-tau species and (e)MTBR-tau243 could potentially serve as an alternative to PET imaging for assessment of continuous AD pathological burden, improving accessibility of AD biomarkers in clinical practice.

## Methods

### Participants
This manuscript used data from the BioFINDER-2 cohort. BioFINDER-2 was approved by the Ethical Review Board in Lund, Sweden, which is part of the Swedish Ethical Review Authority (#2016-1053). We included 784 participants with available plasma biomarker data and 446 with CSF biomarker data from the Swedish BioFINDER-2 study (NCT03174938) across the clinical continuum, ranging from cognitively unimpaired (CU), subjective cognitive decline (SCD), mild cognitive impairment (MCI) to dementia. Participants were recruited at Skåne University Hospital and the Hospital of Ängelholm in Sweden and enrolled between April 2017 and March 2022. Detailed inclusion/exclusion criteria have been previously described[19]. Neurologically and cognitively healthy controls were 40 years or older, did not show cognitive symptoms as assessed by a physician specialized in cognitive disorders, had an MMSE score of 26–30, and did not fulfill the criteria for MCI or dementia according to the DSM-5. Participants with SCD were 40 years or older, required to experience cognitive symptoms (in any cognitive domain), and were considered as CU for the current work, in accordance with the research framework by the NIA-AA[42]. MCI patients were at least 40 years of age, were classified as having cognitive symptoms, performing worse than −1.5 standard deviation in at least one cognitive domain according to age and education-stratified test norms[43], with a Mini-Mental State Examination (MMSE) score at least 24, and not fulfilling the criteria for dementia (major neurocognitive disorder according to Diagnostic and Statistical Manual of Mental Disorders - Fifth Edition, DSM-5[44]). Dementia was classified as fulfilling the DSM-5 criteria for major neurocognitive disorder, with AD dementia requiring an abnormal biomarker for Aβ pathology and clinical presentation probably due to AD, based on medical records, cognitive dementia rating (CDR) assessment, activity of daily living (ADL) assessments using the Function Activities Questionnaire (FAQ), and MRI[45]. Exclusion criteria included the presence of severe somatic disease and current alcohol/substance misuse.

### Global cognition
The modified preclinical Alzheimer cognitive composite (mPACC5) was used as a measure of global cognition. It was calculated based on the previously described PACC5 using MMSE, symbol digit modality (SDMT) and animal fluency, reflective of memory, executive/attention, and verbal function, respectively. As logical memory and the free and cued selective reminding tests were not available in BioFINDER, the ten-word delayed recall task from ADAS–cognition (ADAScog) was used (weighted twice), as previously applied in several studies. The mPACC was thus calculated using z-scores based on the distribution in Aβ-negative CU in the following way: (MMSE + (ADAScog delayed recall × 2) + SDMT + animal fluency)/5[43].

### Fluid biomarkers
Measurement of CSF (%p-tau111, %p-tau153, %p-tau181, %p-tau199, %p-tau202, %p-tau205, %p-tau208, %p-tau217, %p-tau231) and plasma (%p-tau181, %p-tau202, %p-tau205, %p-tau217) phosphorylated tau species occupancy (i.e., p-tau/non-p-tau*100), MTBR-tau243 (CSF), eMTBR-tau243 (plasma), and non-phosphorylated midrange tau (np-tau) variants (np-tau151-155, np-tau181-190, np-tau195-210, np-tau212-221, np-tau226-230) was performed at Washington University using an IP/MS method, as previously described[10,12,20]. CSF levels of Aβ$_{42/40}$ were

measured using the Elecsys platform, as previously described[19]. CSF data were collected at baseline for all participants, while plasma data were collected across several study visits. In this case, PET data closest to study visit and with a maximum time interval of 1 year were used.

Sample measurements and data analysis were performed blinded to the diagnostics of the participants.

### PET image acquisition and processing
Image acquisition and processing details are described previously[19]. In brief, Aβ-PET were acquired on digital GE Discovery MI scanners 90–110 min post injection of ~185 MBq [$^{18}$F]Flutemetamol. Using the whole cerebellum reference region, standardized uptake value ratio (SUVR) in the standard Centiloid target ROI as provided on the GAAIN platform were calculated and subsequently calibrated to the Centiloid scale[46]. Aβ-PET positivity was set at 20 CL, in line with the inclusion criteria of the AHEAD study[2]. Of note, as per the study design of BioFINDER-2, most patients with dementia did not undergo PET imaging, thus the Aβ-PET subsets consisted of mostly CU and MCI individuals.

Tau-PET imaging was performed on the same platform as Aβ-PET, 70–90 min post injection of ~370 MBq [$^{18}$F]RO948. SUVRs were created using the inferior cerebellar cortex as a reference region. A volume-weighted FreeSurfer-based composite temporal meta-ROI was created, reflecting Braak I-IV, with 1.36 SUVR as the cut-point for tau-PET positivity[25]. Tau burden was additionally determined for Braak I–II, III-IV, and V-VI ROIs[47].

### Statistical analysis
All analyses were performed in R version 4.2.2. Demographics were assessed using standard statistics as appropriate.

**Standardization of fluid biomarkers.** Two datasets were independently analyzed. First, CSF and plasma biomarkers were log-transformed. Then, all biomarkers and continuous tau-PET burden in the temporal meta-ROI and Braak regions across timepoints were z-scored based on baseline variation of the CU Aβ-negative group (CSF: $n = 97$, plasma: $n = 116$). For CSF Aβ$_{42/40}$, data were inverted such that higher z-scores related to higher abnormality (for consistency across all biomarkers). For visualization purposes of the plasma biomarkers ($n = 784$), all biomarkers and tau-PET burden were also z-scored based on the whole population, as plasma eMTBR-tau243 has essentially zero variance in CU Aβ-negative individuals. No data were excluded from the analyses.

**Biomarker trajectories against PET chronology.** Biomarker trajectories were investigated and compared using the amyloid- and tau-PET derived chronology as references. To determine when CSF and plasma biomarkers reached significant abnormality relative to amyloid- and tau-PET onset, the time-point of crossing z-score>1.96 was determined, bootstrapped 1000 times using a LOESS model.

This approach was repeated for the stratified analysis for *APOE*-ε4 carriership and biological sex. No statistical method was used to predetermine sample size.

**Interindividual variability.** Next, we aimed to investigate the effect of CSF/plasma dynamics and interindividual variability on biomarker trajectories across AD pathological PET chronology. First, we investigated the behavior of separate p-tau species, i.e., %p-tau, p-tau, and np-tau. Then, as MTBR-tau243 does not have a reference epitope, we corrected the biomarker for the reference np-tau212 (reference epitope of p-tau217) to assess whether its association with tau-PET accumulation across Braak regions improved.

**Longitudinal CSF biomarkers.** Subsequently, using longitudinal CSF assessments, we investigated slope changes in CSF p-tau biomarkers and MTBR-tau243, unnormalized and normalized for np-tau212. Linear

mixed models with random intercept and slope were used to derive individual slopes. Longitudinal plasma data were not available.

**Centimarker comparisons.** Finally, we compared CSF and plasma biomarker trajectories for p-tau species that showed optimal performance across modalities (i.e., %p-tau205, %p-tau217) and (e) MTBR-tau243 against Aβ-PET disease duration by calculating the Centimarker metric[48]. Each individual biomarker Centimarker measure was anchored at the mean of cognitively unimpaired amyloid-negative individuals (=0) and amyloid- and tau-positive dementia patients (=100).

### Reporting summary
Further information on research design is available in the Nature Portfolio Reporting Summary linked to this article.

## Data availability
Anonymized data can be made available upon reasonable request from a qualified academic investigator for the sole purpose of replicating procedures and results presented in the article and as long as data transfer is in agreement with EU legislation on the general data protection regulation which should be regulated in a material transfer agreement. Source data are provided with this paper.

## Code availability
The SILA algorithm is freely available at GitHub (https://github.com/Betthauser-Neuro-Lab/SILA-AD-Biomarker). The R code used for analysis of this work can be made available upon request.

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

## Acknowledgements

We would like to express our gratitude to the research volunteers who participated in the studies from which these data were obtained and their supportive families. The precursor of ¹⁸F-flutemetamol was sponsored by GE Healthcare. The precursor of ¹⁸F-RO948 was provided by Roche. Work at the authors' research center was supported the National Institute of Aging (R01AG083740), European Research Council (ADG-101096455), Alzheimer's Association (ZEN24-1069572, SG-23-1061717), GHR Foundation, Swedish Research Council (2022-00775), ERA PerMed (ERAPERMED2021-184), Knut and Alice Wallenberg foundation (2022-0231), Strategic Research Area MultiPark (Multidisciplinary Research in Parkinson's disease) at Lund University, Swedish Alzheimer Foundation (AF-980907), Swedish Brain Foundation (FO2021-0293), Parkinson foundation of Sweden (1412/22), Cure Alzheimer's fund, Rönström Family Foundation, Konung Gustaf V:s och Drottning Victorias Frimurarestiftelse, Skåne University Hospital Foundation (2020-O000028), Regionalt Forskningsstöd (2022-1259) and Swedish federal government under the ALF agreement (2022-Projekt0080 CSF MTBR-tau243 and p-tau study conducted at Washington University was supported by an Eisai industry grant (PIs K.H. and R.J.B.). Plasma eMTBR-tau243 and p-tau study conducted at Washington University was supported by the Knight ADRC Developmental Grant (PI: K.H.). Resources and effort were supported by the Tracy Family SILQ Center (PI: R.J.B.) established by the Tracy Family, Richard Frimel and Gary Werths, GHR Foundation, David Payne, and the Willman Family brought together by The Foundation for Barnes-Jewish Hospital. The funding sources had no role in the design and conduct of the study; in the collection, analysis, interpretation of the data; or in the preparation, review, or approval of the manuscript.

## Author contributions

L.E.C. performed the statistical analyses and drafted the manuscript; G.S., N.B., R.S., and all authors provided feedback on the manuscript; T.B.J. developed the SILA algorithm and provided feedback on the manuscript; O.S. processed the amyloid- and tau-PET scans; S.P. and N.M.-C. did clinical assessments in BioFINDER-2 and provided feedback on the manuscript; S.J. performed the biomarker analysis; R.O. and O.H. supervised the work. R.J.B. supervised the measurements of CSF and plasma biomarkers; K.H. and R.J.B. developed the CSF MTBR-tau243 and plasma eMTBR-tau243 methods and analyzed them; K.H. and S.E.S. provided feedback on the manuscript; N.R.B. provided feedback on the manuscript.

## Funding

## Competing interests

L.E.C. has acquired research support from GE Healthcare and Springer Healthcare (paid by Eli Lilly), both paid to the institution. Dr. Collij is supported by the MSCA Postdoctoral fellowship (#101108819) and Alzheimer Association Research Fellowship (#23AARF-1029663) grants. G.S. has received speaker fees from Springer and Adium. G.S. received funding from the European Union's Horizon 2020 Research and Innovation Program under Marie Sklodowska-Curie action grant agreement number 101061836, an Alzheimer's Association Research Fellowship (AARF-22-972612), the Brightfocus Foundation (A2024007F), the Alzheimerfonden (AF-980942, AF-994514, AF-1012218), Greta och Johan Kocks research grants and travel grants from the Strategic Research Area MultiPark (Multidisciplinary Research in Parkinson's Disease) at Lund University. K.H. is an Eisai-sponsored voluntary research associate professor at Washington University and has received salary from Eisai. K.H., N.R.B., and R.J.B. may receive income based on technology (methods to detect MTBR-tau isoforms and use thereof) (PCT/US2020/046224) licensed by Washington University to C2N Diagnostics. K.H. and R.J.B. may receive income based on technology (anti-tau MTBR antibodies and methods to detect endogenously cleaved fragments of tau and uses thereof) (PCT/US2023/072738) licensed by Washington University to C2N Diagnostics. T.J.B. reported grants from the National Institutes of Health (NIH) during the conduct of the study and travel funding/fellowship from Alzheimer's Association outside the submitted work. R.S. has received consultancy/speaker fees from Eli Lilly, Novo Nordisk, Roche and Triolab. R.O. has received research funding from European Research Council, ZonMw, NWO, National Institute of Health, Alzheimer Association, Alzheimer Nederland, Stichting Dioraphte, Cure Alzheimer's Fund, Health Holland, ERA PerMed, Alzheimerfonden, and Hjarnfonden (all paid to the institutions). R.O. has received research support from Avid Radiopharmaceuticals, Janssen Research & Development, Roche, Quanterix, and Optina Diagnostics, and has given lectures in symposia sponsored by GE Healthcare. He is an advisory board member for Asceneuron and Bristol Myers Squibb. All the aforementioned have been paid to the institutions. He is an editorial board member of Alzheimer's Research & Therapy and the European Journal of Nuclear Medicine and Molecular Imaging. S.P. has acquired research support (for the institution) from Ki Elements/ADDF and Avid. In the past 2 years, he has received consultancy/speaker fees from BioArtic, Eisai, Lilly, Novo Nordisk, and Roche. S.E.S. has served on scientific advisory boards on biomarker testing and clinical care pathways for Eisai and

Novo Nordisk and has received speaking fees for presentations on biomarker testing from Eisai, Eli Lilly, and Novo Nordisk. N.M.-C. has received funding from the Swedish Research Council (2021-02219), the Swedish Alzheimer Foundation (AF-994229), WASP and DDLS Joint call for research projects (WASP/DDLS22-066), EU Join Programme Neurodegenerative Diseases (2019-03401), Family Rönnström Foundation, Family Berg Foundation, the Swedish federal government under the ALF agreement (2022-Projekt0107), the Swedish Brain Foundation (FO2023-0163), Konung Gustaf V:s och Drottning Victorias Frimurarestiftelse, Johan and Greta Kock Foundation, the Skåne University Hospital Foundation, Regionalt Forskningsstöd. He has received consultancy/speaker fees from Biogen, Owkin, and Merck. R.J.B. is an unpaid scientific advisory board member of Roche and Biogen, and receives research funding from Avid Radiopharmaceuticals, Janssen, Roche or Genentech, Eli Lilly, Eisai, Biogen, AbbVie, Bristol Myers Squibb, and Novartis. O.H. is an employee of Eli Lilly and Lund University. The remaining authors declare no competing interests.
