## [Transparent Peer Review file · Nature Communications]

Trajectories of plasma and CSF MTBR-tau243 and phosphorylated-tau species across the Alzheimer's disease continuum

Corresponding Author: Dr Lyduine Collij

Version 0:

Reviewer comments:

Reviewer #1

(Remarks to the Author)

This manuscript employs a pseudo-temporal framework to model trajectories of candidate fluid biomarkers against amyloid-PET and tau-PET 'chronology, modeled as the estimated time before/after PET positivity. The study is well written and provides a useful organization for how plasma and CSF tau species change in relation to PET-defined amyloid and tau chronology. In particular, the direct comparison of MTBR-tau243 vs. other p-tau measurements and its relationship to amyloid and tau burden in individuals at varying clinical stages carries strong translational significance. Overall, this paper integrates several recently established associations of CSF and plasma biomarkers with AD pathology into a cohesive framework for comparing fluid biomarker dynamics in Alzheimer's disease, though there are several concerns regarding data novelty and interpretation of the results. We believe addressing these concerns would strengthen the manuscript.

1. A primary issue concerns the novelty of the data presented in the manuscript. The introduction states that "it remains to be determined if the recently developed plasma endogenously-cleaved MTBR-tau243 (eMTBRtau243) biomarker shows similar associations with tau PET." However, the authors also cite the first plasma eMTBRtau243 paper (Horie et al. [2025]), which reported trajectories for plasma eMTBRtau243 and other %p-tau measures with tau and amyloid PET in the BIOFINDER-2 cohort. The data presented in this manuscript appear to be drawn from the same cohort and CSF and plasma biomarker datasets as prior work (Horie et al., 2023 and 2025). It would be helpful to explicitly state at the end of the introduction how this current work builds on prior studies, given it appears to primarily reanalyze published datasets.

2. The authors should provide more background to the SILA model, given it is a primary aspect of the methodology that differs from prior work.

3. The abstract should be modified to acknowledge that changes in both %p-tau205 and MTRB-tau243 are associated with tau-positivity onset, with MTBR-tau243 additionally mirroring trajectories of cortical tau-PET burden at later stages.

4. In Figure 1, please provide a brief rationale for how biomarkers were selected for inclusion in panels A and D. I assume this was designed to compare/contrast the strongest markers at various disease stages, but this should be made explicit for the reader. I recommend including %p-tau217 and 205 to Panels B and E to maintain the conceptual theme of the panels (all %p-tau trajectories) as well as preserve consistency with Panels C and F. Furthermore, for all trajectory figures, adding more increments on the y-axis would help compare differences between biomarkers at various disease ages.

5. The authors report that some CSF biomarkers (i.e. %p-tau111) but not others (i.e. %p-tau153) plateau after 10 years of amyloid-PET positivity. How is plateauing defined? The curves in Figure 1C for the two biomarkers mentioned above appear nearly identical. However, the curves for these two markers in Supp Fig. 3 are distinct and more closely match what is written in the text. This needs to be resolved, as it is unclear which figure represents the ground truth. %p-tau181 also exhibits a similar trajectory to %p-tau205 relative to tau-PET in Figure 1D/E, but the authors state that "Only %p-tau205, and even more prominently MTBR-tau243, showed continued increased abnormality after tau-PET onset."

6. That statement on Page 5 that tau-PET models exhibited "similar trajectories to A β -PET chronology" should be modified to convey that the pseudo-temporal ordering of biomarkers was similar across tau and amyloid models, considering that the inflection points for biomarker trajectories relative to estimated disease age differs across models.

7. Figure 3A suggests that MTBR-tau243 could reflect higher stages of amyloid burden. How likely is insoluble tau to be a confounding factor (i.e. would the relationship hold if regressing out tau-PET SUVR)? The conclusion that the slope of MTBR-tau243 was reduced when adjusting for np-tau is difficult to appreciate, as these lines appear largely overlapping for amyloid-PET graph and only modestly deviate (max 0.2 z-score difference) after +10 years from tau-PET positivity.

8. The y-axis values for Figure 4 appear incorrect (z-score range -1 to 1), as no biomarker would surpass an abnormality

threshold of $z > 1.96$. Panel 4A should include mPACC, similar to panel 4C.

9. There also appears to be a much wider range of intercept differences between plasma biomarkers compared to CSF biomarkers – how much of this is related to the difference in standardization procedures for CSF vs. plasma (e.g., normalization to CU neg vs. whole population)? To improve interpretation, the y-axis label for all relevant figures should include a subscript that denotes whether z-scores are based on CU neg or the whole population.

10. Supplemental Figure 9 is missing labels for some biomarkers.

11. In Supplemental Figure 9, it is unclear how to interpret the longer time frame for %p-tau181 predicting amyloid positivity vs. tau positivity. High variability in %p-tau181 is mentioned, but presumably this would apply for the tau-PET prediction as well?

12. The SILA model seems to assume that individual participants progress with similar rates of amyloid and tau accumulation, yet inter-individual factors such as sex and APOE4 are known to influence AD pathophysiology. Sex-stratified and APOE4-stratified analyses are recommended to determine the degree to which these established risk factors alter the estimates of time to amyloid and tau PET positivity onsets for each biomarker. Even though these analyses will have larger confidence intervals due to lower power, a supplemental table that compares when males vs. females and APOE4- vs. APOE4+ reach biomarker abnormality thresholds would be informative and would enhance the novelty of the manuscript.

13. The Discussion suggests that CSF non-phosphorylated mid-region tau could be a late-stage biomarker given a notable ceiling effect in the ratios. It may be helpful to clarify if there may be more utility to combining the ratio and non-phosphorylated tau than simply using the value of the phosphorylated species.

14. The Discussion should note that results from Biofinder-2 may not be broadly generalizable, and that future studies should attempt to assess performance in diverse cohorts encompassing other populations and real-world clinics who may have more co-morbidities.

15. Similarly, the Discussion should include commentary on if the findings are likely to be generalizable to other tracers frequently used in clinical practice and research settings, such as florbetapir, flortaucipir, and MK6240.

16. The authors acknowledge that there are relatively minimal amyloid-PET scans from participants at later clinical stages. Overall, less than half of individuals had multiple amyloid-PET or tau-PET scans – please discuss how this influences the SILA model.

Reviewer #2

(Remarks to the Author)

The manuscript by Collij, et al. characterizes the trajectory of a variety of Alzheimer's disease (AD) biomarkers across the pre-symptomatic and symptomatic landscape of AD. As the AD field moves toward precision medicine, a better understanding of these biomarkers and their temporal relation to each other is vital and this manuscript moves forward our understanding, including in an actionable manner for clinical trial development and ultimately clinical use.

This group of investigators is optimally suited to do this work and have already made many substantive contributions to the field. Strengths of the paper include a well-validated and deeply phenotyped cohort, analysis of multiple biomarkers including a mass spectrometry approach to identify precise phospho-tau epitopes. There are, however, ways that the paper could be improved prior to publications. For example, the authors should make clear this is cross-sectional and not longitudinal data, which is not mentioned until the end of the discussion. Further, it is important to highlight that there is a lack of confirmation with commercially available biomarker tests, as the precise method used for biomarker identification was mass spec and many commercial tests are ELISA-based. Finally, the manuscript is currently written for a fairly narrow and specialized audience of scientists who study tau biomarkers. It would be greatly improved through editing towards accessibility to the more general readership at Nature Communications.

Major points:

1. It should be highlighted early in the manuscript that the work is based on cross-sectional, not longitudinal data. Though referenced at the end of the discussion, the use of the word "trajectory" throughout could be misleading in the absence of a strong framing around the cross-sectionality of the data.

2. The main method used to timestamp the patient data is using the "SILA" method on the PET imaging. For even a neurodegenerative disease-focused audience, more introduction on the validation of this method would provide better justification and confidence in the results.

3. In the discussion, it is important to include discussion of why certain biomarkers show different trajectories, because it can inform us about the biology of the disease. For example, there is evidence to suggest that phosphorylated tau is more durable than an unphosphorylated epitope and that pTau217 is the most resistant to intercellular cleavage of the biomarkers (PMID: 40469052).

4. For the plasma cohort, the PET time estimation may have been different from the plasma collection timepoint. It is not clear what the variability of time is and this should be addressed.

5. In the discussion, the authors could comment on how these biomarker trends could translate to clinical use as there is still wide interindividual variability in the values.

Minor comments:

1. Please define % tau.

2. Line 215-217 "eMTBR-tau243 trajectories closely follows..." has a grammatical error.

Reviewer #3

(Remarks to the Author)

Reviewer #4

(Remarks to the Author)

Reviewer #5

(Remarks to the Author)

In this study, Collij and colleagues investigate the correspondence between leading Alzheimer's disease (AD) fluid biomarkers and PET-based AD pathology quantification. These relationships are examined in a large cohort of individuals across the AD clinical continuum with both cross sectional and longitudinal data, with a focus on plasma and cerebrospinal fluid (CSF) measurements of pTau217 and microtubule binding region (MTBR)-pTau243. Abnormality in these fluid biomarkers is compared with a PET-based threshold for disease onset established using the SILA algorithm.

The authors show that pTau217 exhibits the earliest changes in biomarker abnormality, prior to the onset of PET-based amyloid positivity. Further, they show that MTBR-pTau243 increases in a temporally linked manner with tau PET quantification, and pTau205 abnormality occurs closest to tau PET positivity threshold in disease time. Broad agreement across CSF and plasma measurements of these biomarkers is discussed, as well as interesting dynamics such as ceiling effects due to delayed increases in non-phosphorylated tau species.

Overall, this is a well-conducted and robust study of temporal fluid biomarker dynamics in AD, and is an important contribution to the literature in view of future clinical trials that may use these biomarkers to establish intervention efficacy and timing. I have a few concerns that should be addressed prior to publication:

Concerns:

- Fluid biomarker "abnormality" is discussed but not clearly defined early in the manuscript. The operationalization of values greater than 2SD from the control group should be made clear early on. Can the authors also justify log-transforming and z-scoring fluid biomarker values given that this is not a universal approach in the fluid biomarker literature? Also, the methods and figure legends sometimes confuse 2 SD and a z-score of 1.96 (sometimes 1.96 SD is referenced instead).
- The authors should check all figures and legends for readability and correct labeling (e.g. difficult-to-read labels in Figure 1, lack of panel labels in Supplementary Figure 8, etc.)
- A tau PET threshold of SUVR=1.36 based on prior literature using the Roche tracer is used, but the cortical region this SUVR is calculated from is not made explicit early on (i.e. line 108). The figure legends state this is a metatemporal ROI, but this should be stated in the main text.
- The authors should characterize how much longitudinal data is available for each examined modality. Differences between cross-sectional and longitudinal analyses are discussed, but it is not clear e.g. how many time points there are on average for each plasma, CSF, and PET biomarker.
- The authors should discuss the limitations of the SILA-based assessment of tau PET positivity in terms of 1) the assumption of a single rate of biomarker increase across individuals and 2) the assumption of spatial homogeneity and the use of the meta-temporal ROI as the region where tau PET positivity is quantified.

Minor concerns:

- Line 129: Figure 1C is referenced but appears to discuss data in Figure 1D
- Line 131: the use of the word "linked" is ambiguous, consider changing to something like "abnormality in these two modalities was estimated to occur at approximately the same disease time (~3 years)."
- Line 145: please clarify what is meant by tau PET positivity being closely associated with abnormality in cognition (mPACC abnormality close in time to tau PET positivity?)
- Lines 162-167: this is an important analysis linked MTBR with tau PET, but is written in a slightly confusing way. Can the authors be more clear in stating how this accounts for interindividual variability (in tau PET spatial heterogeneity?)
- Line 173: Please clarify, it's hard to appreciate the claim that pTau205 has a different trajectory than pTau217 in Figure 3A, and the referenced supplementary figure is hard to read and not clearly labeled.
- Lines 197-199: This is also unclear, are the authors claiming that pTau217 continues to increase following tau PET positivity in plasma, but not CSF? What figure shows this dynamic?

Version 1:

Reviewer comments:

Reviewer #2

(Remarks to the Author)

The authors have addressed my concerns.

Reviewer #4

(Remarks to the Author)

Reviewer #5

(Remarks to the Author)

The authors have made considerable improvements from the initial draft in the clarity and limitations of the manuscript. However, there are a few remaining minor concerns that should be addressed prior to publication:

- The authors have made a large number of changes and caveats in response to reviewer feedback. However, a critical limitation of the SILA model approach that is not currently addressed in the manuscript is that biomarkers are assumed to be increasing at the same rate across the population when estimating time to AB/tau positivity. This is an inherent feature of SILA and does not change the fact that these are an interesting set of results, but should be addressed either when introducing the SILA approach or when discussing the limitations of the study.
- Restructuring the results section to begin with a description of the methods has improved the readability of the manuscript. However, can the authors add to this text citations for the abnormality thresholds of 20 CL and 2.36 tau PET SUVR that are included later in the methods section?
- Reviewer #1's point about biomarkers being inconsistently included in the different panels of Figure 1 is valid and may not be fully addressed. I recommend adding the bolded label "Biomarkers of interest" above the legend to the right of Figure 1A and 1D, and then use the note in the figure legend to explain how these biomarkers were selected.
- In response to Reviewer #1's comment 7, please provide a reference for the claim in the updated text: "In the case of amyloid-PET disease duration, this is mostly likely driven by insoluble tau burden, which is known to increase at higher levels of amyloid..."
- The results that were added in response to Reviewer #1's comment 12 are a nice addition to the manuscript. Please note "per se" should be two words in line 339.
- In the limitations paragraph of the discussion section, "sixth" is used twice and should be fixed.

Response to reviewers

Reviewer #1 (Remarks to the Author):

This manuscript employs a pseudo–temporal framework to model trajectories of candidate fluid biomarkers against amyloid-PET and tau-PET ‘chronology, modeled as the estimated time before/after PET positivity. The study is well written and provides a useful organization for how plasma and CSF tau species change in relation to PET-defined amyloid and tau chronology. In particular, the direct comparison of MTBR-tau243 vs. other p-tau measurements and its relationship to amyloid and tau burden in individuals at varying clinical stages carries strong translational significance. Overall, this paper integrates several recently established associations of CSF and plasma biomarkers with AD pathology into a cohesive framework for comparing fluid biomarker dynamics in Alzheimer’s disease, though there are several concerns regarding data novelty and interpretation of the results. We believe addressing these concerns would strengthen the manuscript.

We thank the reviewer for their appreciation of our work and their suggestions to further strengthen the manuscript. Below, they’ll find a point-by-point response to their comments.

1. A primary issue concerns the novelty of the data presented in the manuscript. The introduction states that “it remains to be determined if the recently developed plasma endogenously-cleaved MTBR-tau243 (eMTBRtau243) biomarker shows similar associations with tau PET.” However, the authors also cite the first plasma eMTBRtau243 paper (Horie et al. [2025]), which reported trajectories for plasma eMTBRtau243 and other %p-tau measures with tau and amyloid PET in the BIOFINDER-2 cohort. The data presented in this manuscript appear to be drawn from the same cohort and CSF and plasma biomarker datasets as prior work (Horie et al., 2023 and 2025). It would be helpful to explicitly state at the end of the introduction how this current work builds on prior studies, given it appears to primarily reanalyze published datasets.

We thank the reviewer for this important comment, to better highlight the novelty of our work. Also, the suggestion of investigating the effect of *APOE-ε4* carriership in particular has resulted in interesting additional findings and discussion points.

To clarify our initial position, please note that we are the first study to compare the timing and disease trajectories of this large variety of species (*i.e.*, p-tau species, non-phosphorylated species, and (e)MTBR-tau243) in both CSF and plasma against disease duration rather than pathological burden, as determined by Aβ-PET and tau-PET. Moreover, we uniquely investigated the behavior of non-phosphorylated species, the CSF markers longitudinally, and added the potential effects of *APOE-ε4* carriership and biological sex, though the latter showed no clear effect.

We now highlighted these unique aspects more clearly at the end of the introduction section:

*‘The aim of this study was therefore to investigate the temporal onset of abnormality and trajectories of mass spectrometry-based cross-sectional measurements of MTBR-tau243 and p-tau species in both CSF and plasma against disease time in relation to Aβ-PET and tau-PET positivity, as determined with the SILA algorithm²². In addition, we assessed how these trajectories were influenced by the behavior of the non-phosphorylated tau species and whether they were affected by *APOE-ε4* carriership and biological sex. Finally, longitudinal CSF data was available to investigate changes in biomarker slope against*

disease time. To this end, we included BioFINDER-2 participants across the clinical continuum with available CSF MTBR-tau243 and p-tau and plasma eMTBR-tau243 and p-tau measurements.'

Also, particularly the timing of abnormality onset is novel and key for implementation of these markers in trials and clinical routine, supporting accurate patient selection. We highlighted this aspect to the summarizing paragraph in the Discussion section:

'These findings contribute to our understanding of the trajectories of tau biomarkers in CSF and plasma throughout AD progression, and particularly their timing of abnormality onset, which is key to optimizing their use in clinical practice and clinical trials.'

2. The authors should provide more background to the SILA model, given it is a primary aspect of the methodology that differs from prior work.

We moved the paragraph on the SILA algorithm from the methods to the start of the results section and updated the demographics paragraph to allow for natural flow of the text. Now, key information regarding the SILA algorithm is provided early in the manuscript, which significantly improved the readability. We thank the reviewer for this valuable comment. See below the updated paragraphs of the results section:

Demographics

The CSF cohort (n=446) consisted of 183 cognitively unimpaired participants and 263 cognitively impaired patients (mild cognitive impairment [MCI]: n=124; dementia: n=139), of which 219 (49.1%) were females, 258 (57.8%) were APOE- ϵ 4 carriers, and the mean age at baseline was 71.4 (\pm 8.50) years old (Table-1). A β -PET was available for 261 (58.5%) non-demented subjects, while tau-PET was available for all participants.

The plasma cohort (n=784) consisted of 331 cognitively normal and 453 impaired patients (MCI n=245; dementia: n=208), of which 398 (50.8%) were females, 477 (60.8%) were APOE- ϵ 4 carriers, with a mean age of 72.2 (\pm 9.27) years (Table-1). A β -PET was available for 576 (73.5%) non-demented subjects, while tau-PET was available for all participants.

PET-chronology measure

To retrieve an individual's disease duration, the estimated time of amyloid and tau PET positivity at visit closest to the fluid biomarker was determined by applying the previously developed sampled iterative local approximation (SILA) algorithm to the whole BioFINDER-2 PET dataset. This dataset included A β -PET of 1408 individuals, of which 686 were longitudinal, with an average of 2.33 scans and mean follow-up time 2.96 \pm 1.04 years) and tau-PET of 2003 individuals, of which 922 had longitudinal data, with an average of 2.35 scans and a mean follow-up time 2.83 \pm 1.07 (Supplementary Figure-1)²². Global A β -PET burden was expressed in Centiloid (CL) units using the standard target mask available on the GAIAIN website, while tau-PET burden was expressed in standard uptake value ratio's (SUVRs) in a temporal meta-ROI, reflecting Braak I-IV (see methods). The algorithm uses discrete sampling of CL/SUVR for tau-PET versus age data to establish the relationship between CL/SUVR rate and CL/SUVR. Numerical smoothing (robust LOESS) and Euler's method are used to numerically integrate these data to generate a non-parametric CL/SUVR versus time curve. To give the integrated timeline meaning, the SILA algorithm sets time equal zero to a user-specified value (i.e., threshold, 'tipping point'), which was set at 20 CL and 1.36 SUVR for A β -PET and tau-PET, respectively, to demarcate the zero time corresponding to the A+ and T+ threshold, respectively. The estimated years from biomarker positivity is calculated for each person by first solving this curve for time using a person's observed CL/SUVR, and subtracting the estimated A+ duration from their age at that scan²². This amyloid/tau 'chronology' can be interpreted as the time from PET-detectable amyloid/tau accumulation and serves as the main outcome in the current work.

The SILA algorithm is freely available at GitHub (<https://github.com/Betthausen-Neuro-Lab/SILA-AD-Biomarker>).

3. The abstract should be modified to acknowledge that changes in both %p-tau205 and MTRB-tau243 are associated with tau-positivity onset, with MTBR-tau243 additionally mirroring trajectories of cortical tau-PET burden at later stages.

We thank the reviewer for this comment, and have updated the abstract:

'Changes in %p-tau205 and MTBR-tau243 were closely associated with tau-PET positivity onset, where MTBR-tau243, and especially plasma eMTBR-tau243, was closely associated with trajectories of cortical tau-PET burden at later disease stages.'

4. In Figure 1, please provide a brief rationale for how biomarkers were selected for inclusion in panels A and D. I assume this was designed to compare/contrast the strongest markers at various disease stages, but this should be made explicit for the reader. I recommend including %p-tau217 and 205 to Panels B and E to maintain the conceptual theme of the panels (all %p-tau trajectories) as well as preserve consistency with Panels C and F. Furthermore, for all trajectory figures, adding more increments on the y-axis would help compare differences between biomarkers at various disease ages.

This split was indeed made for readability of the figures. The markers highlighted in panels A and D were those serving as reference standards (i.e. tau-PET burden and mPACC), identified as either the earliest (A β 42/40) or latest changes (np-tau) or showing the largest increment (MTBR-243, ptau205, and ptau217) with increasing PET positivity time, while all other markers showed highly similar behavior or no change (i.e. ptau199 and ptau202) and were therefore clustered in panels B and E.

This consideration has been added to the figure 1 footer:

'Note, biomarkers highlighted in panels A and D included those serving as reference standards (i.e. tau-PET burden and mPACC), identified as either the earliest (A β 42/40) or latest changes (np-tau mid-region) or showing the largest increment (MTBR-tau243, %p-tau205, and %p-tau217) with increasing PET positivity time. Also, %p-tau199 and %p-tau202 are not included in the forest plot, as they did not reach sufficient biomarker abnormality at any point of positivity estimations amyloid-/tau-PET.'

Additional increments on the y-axis have also been added to Figure-1.

Please note that a full figure including all biomarkers together is provided in supplementary Figure 3. As such we opted to not additionally include %p-tau217 and %p-tau205 in panels B and E.

5. The authors report that some CSF biomarkers (i.e. %p-tau111) but not others (i.e. %p-tau153) plateau after 10 years of amyloid-PET positivity. How is plateauing defined? The curves in Figure 1C for the two biomarkers mentioned above appear nearly identical. However, the curves for these two markers in Supp Fig. 3 are distinct and more closely match what is written in the text. This needs to be resolved, as it is unclear which figure represents the ground truth. %p-tau181 also exhibits a similar trajectory to %p-tau205 relative to tau-PET in Figure 1D/E, but the authors state that "Only %p-tau205, and even more prominently MTBR-tau243, showed continued increased abnormality after tau-PET onset."

Apologies for this typo and error in supplementary Figure 1. The reviewer is correct that %p-tau153 also demonstrates a plateauing effect, showing a highly similar trajectory to %p-tau181. The updated supplementary figure now demonstrates differences in %p-tau181 and

%p-tau205 trajectories, in line with Figure 1 of the main text (see below updated supplementary Figure 1).

Regarding the definition of plateauing, curve characteristics were assessed through visual inspection, while abnormality onset was determined through bootstrapped LOESS models. We now added throughout the results section when visual inspection was applied.

6. That statement on Page 5 that tau-PET models exhibited “similar trajectories to Aβ-PET chronology” should be modified to convey that the pseudo-temporal ordering of biomarkers was similar across tau and amyloid models, considering that the inflection points for biomarker trajectories relative to estimated disease age differs across models.

Wording has been updated to: *‘The pseudo-temporal ordering of biomarkers was similar to Aβ-PET chronology models...’*

7. Figure 3A suggests that MTBR-tau243 could reflect higher stages of amyloid burden. How likely is insoluble tau to be a confounding factor (i.e. would the relationship hold if regressing out tau-PET SUVR)? The conclusion that the slope of MTBR-tau243 was reduced when adjusting for np-tau is difficult to appreciate, as these lines appear largely overlapping for amyloid-PET graph and only modestly deviate (max 0.2 z-score difference) after +10 years from tau-PET positivity.

We thank the reviewer for this interesting question. Indeed, MTBR-tau243 has been shown to be particularly determined by insoluble tau, which is most likely occurring in those individuals with longer amyloid-PET positivity duration or higher amyloid load. Indeed, correlation analysis between the two time measures illustrates that those with longer amyloid-PET positivity duration, also tend to have longer tau-PET positivity duration, see figure below.

Regarding the comment of reduced MTBR-tau243 slope, this conclusion was mainly derived based on Figure 3 panel B, as panel A includes only non-demented individuals (as amyloid-PET was not available for patients with dementia in the BioFINDER-2 study), which have lower levels of tau-PET burden, and indeed the effect is not observed. Instead, the lower slope of MTBR-tau243 can only be observed upon (very) high tau burden, reflected in panel B.

We updated the results text to clarify these findings:

‘Similarly, MTBR-tau243 demonstrated increased slopes upon high AD pathological PET chronology. In case of amyloid-PET disease duration (Figure-3A), this is most likely driven by insoluble tau burden, which is known to increase at higher levels of amyloid and given the identical trajectory observed against tau-PET disease duration (Figure-3B). The slope of MTBR-tau243 was reduced when adjusting the marker for the reference epitope np-tau212, mainly at high tau-PET disease duration. This is due to the increased accumulation of np-tau212 in individuals with higher tau burden (Figure-3B).’

8. The y-axis values for Figure 4 appear incorrect (z-score range -1 to 1), as no biomarker would surpass an abnormality threshold of $z > 1.96$. Panel 4A should include mPACC, similar to panel 4C.

We apologize for the confusion. In contrast to CSF, z-scores Figures 4 panels A and C for plasma are based on the whole population, rather than based on a reference CN A β -population, as plasma eMTBR-tau243 has essentially zero variance in CU A β -negative individuals and visualization using a reference group z-score approach was not feasible. This is stated in the statistical analysis section, but for clarification, we added this information to the y-axis of the figure and the figure footer:

‘A/C illustrate changes in z-scores, based on the mean and standard deviation of the whole population, rather than a CU A β - reference group, as plasma eMTBR-tau243 has essentially zero variance in CU A β -negative individuals. Shaded area’s indicate the 95% confidence interval.’

9. There also appears to be a much wider range of intercept differences between plasma biomarkers compared to CSF biomarkers – how much of this is related to the difference in standardization procedures for CSF vs. plasma (e.g., normalization to CU neg vs. whole population)? To improve interpretation, the y-axis label for all relevant figures should include a subscript that denotes whether z-scores are based on CU neg or the whole population.

In line with the comment above, we further clarified this difference for plasma biomarker visualization in Figure 4 by updated the y-axis label, which now includes that z-scoring was based on the whole-population.

10. Supplemental Figure 9 is missing labels for some biomarkers.

Apologies for this oversight. The labels have been added.

11. In Supplemental Figure 9, it is unclear how to interpret the longer time frame for %p-tau181 predicting amyloid positivity vs. tau positivity. High variability in %p-tau181 is mentioned, but presumably this would apply for the tau-PET prediction as well?

This is probably the result of a combination of the permutation testing, which can yield slightly different confidence intervals and the fact that around this disease stage more individuals with tau-PET are available than with amyloid-PET, as patients with dementia do not have amyloid-PET scans.

12. The SILA model seems to assume that individual participants progress with similar rates of amyloid and tau accumulation, yet inter-individual factors such as sex and APOE4 are known to influence AD pathophysiology. Sex-stratified and APOE4-stratified analyses are recommended to determine the degree to which these established risk factors alter the estimates of time to amyloid and tau PET positivity onsets for each biomarker. Even though these analyses will have larger confidence intervals due to lower power, a supplemental table that compares when males vs. females and APOE4- vs. APOE4+ reach biomarker abnormality thresholds would be informative and would enhance the novelty of the manuscript.

We thank the reviewer for this valuable comment. Stratified analysis for biological sex and APOE-ε4 carriership have been added to the results section on the manuscript. For CSF, these effects were investigated for both Aβ-PET and tau-PET trajectories. For plasma, only for tau-PET, as Aβ-PET trajectories for determining significant abnormality onset were less informative in this dataset.

In general, we observed no effect of biological sex or APOE-ε4 carriership on the general order of CSF biomarkers reaching significant abnormality. Across all strata, %p-tau217 consistently emerged as the earliest biomarker to become abnormal, %p-tau205 and MTBR-tau243 to closely align with tau-PET onset, and finally np-tau mid-region showing abnormality late in the disease course.

However, APOE-ε4 carriers reached abnormality for all CSF biomarkers earlier than non-carriers, relative to both Aβ-PET and tau-PET reference onsets, though based on the 95% confidence intervals, this difference was only significant for %p-tau217 and %p-tau181 relative to Aβ-PET and tau-PET chronicity, and MTBR-tau243 relative to tau-PET chronicity.

A similar observation was made for plasma biomarkers, with the general ordering not affected by either biological sex or APOE-ε4. Nonetheless, APOE-ε4 carriers showed earlier changes in %p-tau217, %p-tau205, and mPACC compared to non-carriers, though not for MTBR-tau243. Note, these differences did not reach statistical significance given the wider confidence intervals in these stratified analyses due to the smaller sample sizes of the subpopulations.

These findings are now described in the results section and full findings can be found in **Supplementary Table-1** and **Supplementary Table-2**.

In addition, we added a paragraph in the discussion regarding the APOE-ε4 findings:

‘Our stratified analysis suggests that APOE-ε4 carriers demonstrated changes in both CSF and plasma biomarkers earlier than non-carriers, though not a shorter disease trajectory perse. As the SILA model is anchored in the same positivity cut-point for both subpopulations, and previous work has illustrated that the rate of Aβ-PET accumulation is not affected by APOE-ε4 carriership²², this difference is not driven by the presence of higher aggregated Aβ, as measured by PET, in APOE-ε4 carriers. Instead, a potential underlying biological mechanism could be that fluid and PET Aβ provide partially independent information³⁴ and APOE-ε4 seems to affect the balance between soluble and aggregated Aβ forms. In particular, APOE-ε4 was found to promote the accumulation of amorphous Aβ assemblies, which are more reflected in the fluid biomarkers compared to the PET imaging³⁵. In line, Mastenbroek and colleagues (2024) reported that homozygous APOE-ε4 carriers had relatively more soluble Aβ compared to fibrillar Aβ³⁶. Together with the observation that most p-tau forms reflect both Aβ and tau pathology³⁷, it is possible that this shift in biomarker abnormality onset for APOE-ε4 carriers reflects the relatively higher presence of soluble pathological markers in this subpopulation, making them potentially more sensitive for early pathological detection than in APOE-ε4 non-carriers. Given that these analyses were performed in subpopulations resulting in much larger confidence intervals, these findings should be interpreted with caution and be validated in an external dataset. Related, due to cohort size, we were not able to investigate the effect of APOE-ε4 within males and females separately. As previous studies reported faster disease progression for particularly female APOE-ε4 carriers³⁸, it would be of interest to investigate the potential interplay between APOE-ε4 carriership and biological sex on biomarker disease trajectories.’

13. The Discussion suggests that CSF non-phosphorylated mid-region tau could be a late-stage biomarker given a notable ceiling effect in the ratios. It may be helpful to clarify if there may be more utility to combining the ratio and non-phosphorylated tau than simply using the value of the phosphorylated species.

The conclusion paragraph of the discussion has been adapted to reflect this consideration: *‘Taken together, concurrent assessment of (e)MTBR-tau243, p-tau217, p-tau205, and potentially phosphorylated mid-region tau can accurately track PET-based AD pathological burden. This can support staging efforts at the individual level using CSF or plasma measurements, which is an increasing need for clinical trial design.’*

14. The Discussion should note that results from Biofinder-2 may not be broadly generalizable, and that future studies should attempt to assess performance in diverse cohorts encompassing other populations and real-world clinics who may have more comorbidities.

We thank the reviewer for this important point. We added the following consideration to the discussion section:

‘Finally, while the BioFINDER-2 cohort consists of a memory clinic population, it may not be broadly generalizable, and future studies should include larger more real-world populations, including cohorts with a higher frequency of other neurodegenerative or psychiatric diseases, medical comorbidities and other demographic backgrounds that may affect biomarker, and particularly plasma biomarker, measurements.’

15. Similarly, the Discussion should include commentary on if the findings are likely to be generalizable to other tracers frequently used in clinical practice and research settings, such as florbetapir, flortaucipir, and MK6240.

We thank the reviewer for this comment and added the following methodological consideration to the discussion section:

'Third, this study utilized the radiotracers [¹⁸F]flutemetamol and [¹⁸F]RO948 for assessment of A β and tau burden, respectively. Though amyloid-PET was harmonized using the Centiloid approach and our model fit was highly consistent with previous reports using the SILA method on other F-18 radiotracers, such as [¹⁸F]florbetapir³³, harmonization for tau-PET tracers is still not established and differences in tau-PET tracer kinetics might affect their association with fluid biomarker trajectories. Nonetheless, a previous head-to-head study from our group has shown highly comparable performance between [¹⁸F]RO948 and the FDA-approved [¹⁸F]flortaucipir tracer³⁴.'

16. The authors acknowledge that there are relatively minimal amyloid-PET scans from participants at later clinical stages. Overall, less than half of individuals had multiple amyloid-PET or tau-PET scans – please discuss how this influences the SILA model.

The reviewer is correct that amyloid-PET was not available in patients with dementia, though a wide range of amyloid burden was observed due to the inclusion of MCI individuals, the disease stage where this biomarker tends to plateau. This can also be appreciated in Supplementary Figure 1 (see below amyloid panels). As such, the amyloid-PET SILA model was not affected by limited high amyloid-PET burden data.

In addition, the number of available longitudinal data in the BioFINDER-2 cohort was in fact substantially larger than in most previously reported studies using the SILA algorithm. See for example: https://alz-journals.onlinelibrary.wiley.com/doi/full/10.1002/alz.092486?casa_token=787ptdmIYP8AAAAA%3AQSVcmUJNcfBYSjCBLUFhtBWM2WuqTlue5AuCEHoboQP_eO3-L51P8x1F4NjX4WRUpC1jermoZojE

In addition, our mean follow-up period of 3 years is also a strength, in line with the original SILA paper demonstrating the model fit for amyloid-PET:

<https://academic.oup.com/brain/article/145/11/4065/6646758>

Finally, in line with the response to the previous comment, the model fit was highly comparable to other previous studies utilizing the SILA method.

We expanded the methodological considerations in the Discussion section to reflect the strength of this study in this aspect.

'Fourth, as per BioFINDER-2 study design, A β -PET was only available in very few patients with a dementia diagnosis at baseline, though the whole range of A β burden was well represented due to the inclusion of individuals with MCI and did not affect SILA model fit. A strength of the study was the availability of a relatively large longitudinal PET data-set ($N_{A\beta}$ =686 and N_{τ} =922) compared to previous studies^{22, 33} with an average follow-up time of nearly 3 years for both PET tracers, which supports accurate model fit by the SILA

algorithm.'

Reviewer #2 (Remarks to the Author):

The manuscript by Collij, et al. characterizes the trajectory of a variety of Alzheimer's disease (AD) biomarkers across the pre-symptomatic and symptomatic landscape of AD. As the AD field moves toward precision medicine, a better understanding of these biomarkers and their temporal relation to each other is vital and this manuscript moves forward our understanding, including in an actionable manner for clinical trial development and ultimately clinical use.

This group of investigators is optimally suited to do this work and have already made many substantive contributions to the field. Strengths of the paper include a well-validated and deeply phenotyped cohort, analysis of multiple biomarkers including a mass spectrometry approach to identify precise phospho-tau epitopes. There are, however, ways that the paper could be improved prior to publications. For example, the authors should make clear this is cross-sectional and not longitudinal data, which is not mentioned until the end of the discussion. Further, it is important to highlight that there is a lack of confirmation with commercially available biomarker tests, as the precise method used for biomarker identification was mass spec and many commercial tests are ELISA-based. Finally, the manuscript is currently written for a fairly narrow and specialized audience of scientists who study tau biomarkers. It would be greatly improved through editing towards accessibility to the more general readership at Nature Communications.

We thank the reviewer for their appreciation of our work and their suggestions to further clarify the manuscript. Below, they'll find a point-by-point response to their comments.

Major points:

1. It should be highlighted early in the manuscript that the work is based on cross-sectional, not longitudinal data. Though referenced at the end of the discussion, the use of the word "trajectory" throughout could be misleading in the absence of a strong framing around the cross-sectionality of the data.

We appreciate this important comment from the reviewer. We now have explicitly stated the cross-sectional nature of the biomarker data in the abstract and in the final paragraph on the introduction/aim paragraph:

'The aim of this study was therefore to investigate the temporal onset of abnormality and trajectories of mass spectrometry-based cross-sectional measurements of MTBR-tau243 and p-tau species in both CSF and plasma against disease time in relation to A β -PET and tau-PET positivity, as determined with the SILA algorithm²². In addition, we assessed how these trajectories were influenced by the behavior of the non-phosphorylated tau species and whether they were affected by APOE- ϵ 4 carriership and biological sex. Finally, longitudinal CSF data was available to investigate changes in biomarker slope against disease time. To this end, we included BioFINDER-2 participants across the clinical continuum with available CSF MTBR-tau243 and p-tau and plasma eMTBR-tau243 and p-tau measurements.'

2. The main method used to timestamp the patient data is using the "SILA" method on the PET imaging. For even a neurodegenerative disease-focused audience, more introduction on the validation of this method would provide better justification and confidence in the results.

In line with a comment from Reviewer 1, we now included most of the SILA algorithm methods in the results section, to allow for better flow of the text and validation of the utilized algorithm early on in the manuscript:

'PET-chronology measure

'To retrieve an individual's disease duration, the estimated time of amyloid and tau PET positivity at visit closest to the fluid biomarker was determined by applying the previously developed sampled iterative local approximation (SILA) algorithm to the whole BioFINDER-2 PET dataset. This dataset included A β -PET of 1408 individuals, of which 686 were longitudinal, with an average of 2.33 scans and mean follow-up time 2.96 \pm 1.04 years) and tau-PET of 2003 individuals, of which 922 had longitudinal data, with an average of 2.35 scans and a mean follow-up time 2.83 \pm 1.07 (Supplementary Figure-1)²². Global A β -PET burden was expressed in Centiloid (CL) units using the standard target mask available on the GAAIN website, while tau-PET burden was expressed in standard uptake value ratio's (SUVRs) in a temporal meta-ROI, reflecting Braak I-IV (see methods). The algorithm uses discrete sampling of CL/SUVR for tau-PET versus age data to establish the relationship between CL/SUVR rate and CL/SUVR. Numerical smoothing (robust LOESS) and Euler's method are used to numerically integrate these data to generate a non-parametric CL/SUVR versus time curve. To give the integrated timeline meaning, the SILA algorithm sets time equal zero to a user-specified value (i.e., threshold, 'tipping point'), which was set at 20 CL and 1.36 SUVR for A β -PET and tau-PET, respectively, to demarcate the zero time corresponding to the A+ and T+ threshold, respectively. The estimated years from biomarker positivity is calculated for each person by first solving this curve for time using a person's observed CL/SUVR, and subtracting the estimated A+ duration from their age at that scan²². This amyloid/tau 'chronology' can be interpreted as the time from PET-detectable amyloid/tau accumulation and serves as the main outcome in the current work.

The SILA algorithm is freely available at GitHub (<https://github.com/Betthausen-Neuro-Lab/SILA-AD-Biomarker>).'

In addition, several methodological considerations regarding the SILA algorithm and our particular dataset have been added to the Discussion section, also in line with comments from reviewer 1:

'Third, this study utilized the radiotracers [¹⁸F]flutemetamol and [¹⁸F]RO948 for assessment of A β and tau burden, respectively. Though amyloid-PET was harmonized using the Centiloid approach and our model fit was highly consistent with previous reports using the SILA method on other F-18 radiotracers, such as [¹⁸F]florbetapir³³, harmonization for tau-PET tracers is still not established and differences in tau-PET tracer kinetics might affect their association with fluid biomarker trajectories. Nonetheless, a previous head-to-head study from our group has shown highly comparable performance between [¹⁸F]RO948 and the FDA-approved [¹⁸F]flortaucipir tracer³⁴.' Fourth, as per BioFINDER-2 study design, A β -PET was only available in very few patients with a dementia diagnosis at baseline, though the whole range of A β burden was well represented due to the inclusion of individuals with MCI and did not affect SILA model fit. A strength of the study was the availability of a relatively large longitudinal PET data-set ($N_{A\beta}$ =686 and N_{τ} =922) compared to previous studies^{22, 33} with an average follow-up time of nearly 3 years for both PET tracers, which supports accurate model fit by the SILA algorithm.'

3. In the discussion, it is important to include discussion of why certain biomarkers show different trajectories, because it can inform us about the biology of the disease. For example, there is evidence to suggest that phosphorylated tau is more durable than an unphosphorylated epitope and that pTau217 is the most resistant to intercellular cleavage of the biomarkers (PMID: 40469052).

Thank you for providing us with this interesting reference! We have added the consideration to the discussion section:

'For example, %p-tau217 outperformed %p-tau181 for both CSF and plasma biomarkers, probably due to the larger variability of the latter and the previously described durability of the former. More specifically, previous work has illustrated that %p-tau217 is more resistant against intracellular cleavage, meaning that even a modest early increase in the molecule can accumulate into a detectable signal²⁹. These findings are also in line with previous literature, which reported optimal performance of %p-tau217 to predict A β -status^{13, 15, 28}.'

4. For the plasma cohort, the PET time estimation may have been different from the plasma collection timepoint. It is not clear what the variability of time is and this should be addressed.

We apologize for this missing information. We added to the methods section that PET data closest to plasma data collection, with a maximum time interval of 1 year was used for subsequent analysis:

'CSF data were collected at baseline for all participants, while plasma data were collected across several study visits. In this case, PET data closest to study visit and with a maximum time interval of 1 year was used.'

5. In the discussion, the authors could comment on how these biomarker trends could translate to clinical use as there is still wide interindividual variability in the values.

We highlighted the potential clinical use of combining these biomarkers further in the end of the Discussion section:

'In addition, combining different p-tau species and (e)MTBR-tau243 could potentially serve as an alternative to PET imaging for assessment of continuous AD pathological burden, improving accessibility of AD biomarkers in clinical practice.'

Minor comments:

1. Please define % tau. This is now specified on first mentioning in the results section: *'(i.e., phosphorylated tau species occupancy or p-tau/non-p-tau*100)'*
2. Line 215-217 "eMTBR-tau243 trajectories closely follows..." has a grammatical error. Corrected, thank you for spotting!

Reviewer #3 and #4 (Remarks to the Author):

Reviewer #5 (Remarks to the Author):

In this study, Collij and colleagues investigate the correspondence between leading Alzheimer's disease (AD) fluid biomarkers and PET-based AD pathology quantification. These relationships are examined in a large cohort of individuals across the AD clinical continuum with both cross sectional and longitudinal data, with a focus on plasma and cerebrospinal fluid (CSF) measurements of pTau217 and microtubule binding region

(MTBR)-pTau243. Abnormality in these fluid biomarkers is compared with a PET-based threshold for disease onset established using the SILA algorithm.

The authors show that pTau217 exhibits the earliest changes in biomarker abnormality, prior to the onset of PET-based amyloid positivity. Further, they show that MTBR-pTau243 increases in a temporally linked manner with tau PET quantification, and pTau205 abnormality occurs closest to tau PET positivity threshold in disease time. Broad agreement across CSF and plasma measurements of these biomarkers is discussed, as well as interesting dynamics such as ceiling effects due to delayed increases in non-phosphorylated tau species.

Overall, this is a well-conducted and robust study of temporal fluid biomarker dynamics in AD, and is an important contribution to the literature in view of future clinical trials that may use these biomarkers to establish intervention efficacy and timing. I have a few concerns that should be addressed prior to publication:

We thank the reviewers for their appreciation of our work and their suggestions to further strengthen the manuscript. Below, they'll find a point-by-point response to their comments.

Concerns:

- Fluid biomarker “abnormality” is discussed but not clearly defined early in the manuscript. The operationalization of values greater than 2SD from the control group should be made clear early on. Can the authors also justify log-transforming and z-scoring fluid biomarker values given that this is not a universal approach in the fluid biomarker literature? Also, the methods and figure legends sometimes confuse 2 SD and a z-score of 1.96 (sometimes 1.96 SD is referenced instead).

In line with comments from reviewers 2 and 3, we now moved the majority of the methods regarding the SILA algorithm and definition of PET abnormality to the results section to allow for better flow of the text and context of the subsequent main results:

'PET-chronology measure

To retrieve an individual's disease duration, the estimated time of amyloid and tau PET positivity at visit closest to the fluid biomarker was determined by applying the previously developed sampled iterative local approximation (SILA) algorithm to the whole BioFINDER-2 PET dataset. This dataset included A β -PET of 1408 individuals, of which 686 were longitudinal, with an average of 2.33 scans and mean follow-up time 2.96 \pm 1.04 years) and tau-PET of 2003 individuals, of which 922 had longitudinal data, with an average of 2.35 scans and a mean follow-up time 2.83 \pm 1.07 (Supplementary Figure-1)22. Global A β -PET burden was expressed in Centiloid (CL) units using the standard target mask available on the GAAIN website, while tau-PET burden was expressed in standard uptake value ratio's (SUVRs) in a temporal meta-ROI, reflecting Braak I-IV (see methods). The algorithm uses discrete sampling of CL/SUVR for tau-PET versus age data to establish the relationship between CL/SUVR rate and CL/SUVR. Numerical smoothing (robust LOESS) and Euler's method are used to numerically integrate these data to generate a non-parametric CL/SUVR versus time curve. To give the integrated timeline meaning, the SILA algorithm sets time equal zero to a user-specified value (i.e., threshold, 'tipping point'), which was set at 20 CL and 1.36 SUVR for A β -PET and tau-PET, respectively, to demarcate the zero time corresponding to the A+ and T+ threshold, respectively. The estimated years from biomarker positivity is calculated for each person by first solving this curve for time using a person's observed CL/SUVR, and subtracting the estimated A+ duration from their age at that scan22. This amyloid/tau 'chronology' can be interpreted as the time from PET-detectable amyloid/tau accumulation and serves as the main outcome in the current work.

The SILA algorithm is freely available at GitHub (<https://github.com/Betthausen-Neuro-Lab/SILA-AD-Biomarker>).

Also, we now specify the definition of abnormality at first mentioning in the results section:

'The only tau biomarker that reached significant abnormality (i.e., reaching >1.96 z-score compared to the reference population) before A β -PET positivity was %p-tau217, which became abnormal 0.6 year (95% CI 1.7–0.4) prior to A β -PET positivity (Figure-1C).'

Thank you for noticing the wrong use of 2SD in the footer of Figure 1. This has been adapted.

- The authors should check all figures and legends for readability and correct labeling (e.g. difficult-to-read labels in Figure 1, lack of panel labels in Supplementary Figure 8, etc.)

We apologies for any errors. Figures have been updated accordingly.

- A tau PET threshold of SUVR=1.36 based on prior literature using the Roche tracer is used, but the cortical region this SUVR is calculated from is not made explicit early on (i.e. line 108). The figure legends state this is a metatemporal ROI, but this should be stated in the main text.

Thank you for raising this point to clarify the ROIs early on. We added the following text to the new 'PET chronology-measure' paragraph in the results section:

'Global A β -PET burden was expressed in Centiloid (CL) units using the standard target mask available on the GAAIN website, while tau-PET burden was expressed in standard uptake value ratio's (SUVRs) in a temporal meta-ROI, reflecting Braak I-IV (see methods).'

- The authors should characterize how much longitudinal data is available for each examined modality. Differences between cross-sectional and longitudinal analyses are discussed, but it is not clear e.g. how many time points there are on average for each plasma, CSF, and PET biomarker.

We appreciate the comment, as this information is currently found in the methods section only. We added the relevant info to the results section for CSF and PET, as plasma was not available longitudinally:

*'SILA modelled data based on 1408 individuals with A β -PET, of which 686 were longitudinal, with an average of 2.33 scans and mean follow-up time 2.96 \pm 1.04 years) and 2003 individuals with tau-PET, of which 922 had longitudinal data, with an average of 2.35 scans and a mean follow-up time 2.83 \pm 1.07 can be found in **Supplementary Figure-1.***

'Longitudinal CSF data was available for 218 individuals, with a mean follow-up time of 2.04 years (SD= 0.21, range: 1.63-3.06 years).'

- The authors should discuss the limitations of the SILA-based assessment of tau PET positivity in terms of 1) the assumption of a single rate of biomarker increase across individuals and 2) the assumption of spatial homogeneity and the use of the meta-temporal ROI as the region where tau PET positivity is quantified.

Thank you for raising this interesting point. We added the following future research consideration to the discussion section:

'Fifth, this study utilized a previously validated and commonly used temporal meta-ROI to assess global tau-PET burden, though known heterogeneity in tau-PET exists³⁵. As such, it would be of interest to investigate whether the investigated fluid biomarkers in this work demonstrate distinct trajectories based on previously described tau-PET subtypes.'

Minor concerns:

- Line 129: Figure 1C is referenced but appears to discuss data in Figure 1D
1C is correct, as it is referencing time to amyloid-PET onset, while 1D is related to tau-PET.

- Line 131: the use of the word “linked” is ambiguous, consider changing to something like “abnormality in these two modalities was estimated to occur at approximately the same disease time (~3 years).”

Wording has been adapted to 'similarly to abnormality in global cognition...'

- Line 145: please clarify what is meant by tau PET positivity being closely associated with abnormality in cognition (mPACC abnormality close in time to tau PET positivity?)

Correct, the sentence has been clarified. Together with specifying 'abnormality' early on in the manuscript, we believe this statement is now clear.

- Lines 162-167: this is an important analysis linked MTBR with tau PET, but is written in a slightly confusing way. Can the authors be more clear in stating how this accounts for interindividual variability (in tau PET spatial heterogeneity?)

Unfortunately, the SILA algorithm deployed in this work does not take spatial heterogeneity into account. In fact, the use of the temporal meta-ROI to determine tau-PET disease duration could be viewed as a limitation from that perspective, though the majority of cases in this cohort are known to have a more classic tau distribution. In line with the comment above, we added the following future research suggestion in the discussion section:

'Fifth, this study utilized a previously validated and commonly used temporal meta-ROI to assess global tau-PET burden, though known heterogeneity in tau-PET exists³⁵. As such, it would be of interest to investigate whether the investigated fluid biomarkers in this work demonstrate distinct trajectories based on previously described tau-PET subtypes.'

- Line 173: Please clarify, it's hard to appreciate the claim that pTau205 has a different trajectory than pTau217 in Figure 3A, and the referenced supplementary figure is hard to read and not clearly labeled.

In figure 3A, it can be appreciated, particularly in the right figure, that %p-tau205 shows more continued accumulation than %p-tau217. We now specify the right figure in the text.

In addition, supplementary figure 8 has been updated to demonstrate sufficient quality.

• Lines 197-199: This is also unclear, are the authors claiming that pTau217 continues to increase following tau PET positivity in plasma, but not CSF? What figure shows this dynamic? We have now specified the figures in which this phenomenon can be appreciated. The continuous increase for plasma can be seen in figure 4D, while the attenuation effect for its CSF counterpart can be appreciated in Figure 1D.

Response to reviewers R2

Reviewer #5 (Remarks to the Author):

The authors have made considerable improvements from the initial draft in the clarity and limitations of the manuscript.

We thank the reviewer for their appreciation of our implemented changes. Please find below a point-by-point response to remaining queries.

However, there are a few remaining minor concerns that should be addressed prior to publication:

- The authors have made a large number of changes and caveats in response to reviewer feedback. However, a critical limitation of the SILA model approach that is not currently addressed in the manuscript is that biomarkers are assumed to be increasing at the same rate across the population when estimating time to AB/tau positivity. This is an inherent feature of SILA and does not change the fact that these are an interesting set of results, but should be addressed either when introducing the SILA approach or when discussing the limitations of the study.

Thank you for raising this important point. The reviewer is correct that this is an assumption underlying the SILA algorithm and has since also been tested extensively to understand its potential limitation. For amyloid PET, this assumption was tested with three different models (SILA, ODE-GP, and GBTM) in three cohorts with sporadic AD (ADNI, WRAP, BLSA)¹. Importantly, no effect for sex, APOE genotype, age, and clinical diagnosis was observed, in line with the findings from the post-hoc analysis in this work. Only in dominantly inherited AD or other genetic AD forms (e.g., Down Syndrome) different trajectories have been observed². Similar work is also ongoing for tau-PET, though not yet published, the developer of the model and co-author of this paper, Dr. Tobey Betthausen, demonstrated that also the temporal META-ROI trajectories, as used in the current work, were not impacted by sex, APOE, age, and diagnosis, similar to amyloid-PET.

We have added these considerations to the Discussion section:

'Sixth, the SILA algorithm assumes individuals follow consistent accumulation trajectories, which could be a limitation. Nonetheless, this assumption has been extensively tested for amyloid PET in sporadic AD, showing no effect of sex, APOE genotype, age, and clinical diagnosis on trajectories. Similar work is currently also ongoing for tau-PET.'

- Restructuring the results section to begin with a description of the methods has improved the readability of the manuscript. However, can the authors add to this text citations for the abnormality thresholds of 20 CL and 2.36 tau PET SUVR that are included later in the methods section?

Relevant references have been added to the results section as well.

- Reviewer #1's point about biomarkers being inconsistently included in the different panels of Figure 1 is valid and may not be fully addressed. I recommend adding the bolded label "Biomarkers of interest" above the legend to the right of Figure 1A and 1D, and then use the note in the figure legend to explain how these biomarkers were selected.

Figure 1 has been updated as suggested. We also added 'Other biomarkers' to panel B and E.

- In response to Reviewer #1's comment 7, please provide a reference for the claim in the updated text: "In the case of amyloid-PET disease duration, this is mostly likely driven by insoluble tau burden, which is known to increase at higher levels of amyloid..."

A key reference for this statement is Collij LE, Bollack A, La Joie R, et al. Centiloid recommendations for clinical context-of-use from the AMYPAD consortium. *Alzheimer's Dement.* 2024;20:9037–9048. <https://doi.org/10.1002/alz.14336>

In this review, the authors summarized the existing literature regarding Centiloid quantification and cut-points and illustrated that cortical tau (PET) burden emerges around 36-50 CL and high tau burden (as defined in the donanemab trials) was observed around 60 CL. The 50 CL mark corresponds with around 8 years of A β -PET positivity in the current work (see Supplementary figure 1). So the changes in MTBR-tau243 slopes occur mainly in the timeframe where high CL and therefore cortical tau burden is apparent.

The reference has been added to the manuscript.

- The results that were added in response to Reviewer #1's comment 12 are a nice addition to the manuscript. Please note "per se" should be two words in line 339.

Thank you, we corrected the typo.

- In the limitations paragraph of the discussion section, "sixth" is used twice and should be fixed.

Counting has been updated also including additional SILA methodological consideration

References

1. Betthauser TJ, Bilgel M, Kosciak RL, et al. Multi-method investigation of factors influencing amyloid onset and impairment in three cohorts. *Brain* 2022;145:4065-4079.
2. Wisch JK, McKay NS, Zammit M, et al. Comparison of amyloid chronicity and EYO in autosomal dominant Alzheimer's disease. *Alzheimer's & Dementia* 2025;21:e70812.